# No-regret Online Learning over Riemannian Manifolds

**Xi Wang**
AMSS
Chinese Academy of Sciences
Beijing, China
wangxi14@mails.ucas.ac.cn

**Zhipeng Tu**
AMSS
Chinese Academy of Sciences
Beijing, China
tuzhipeng@amss.ac.cn

**Yiguang Hong**[*]
Tongji University
Shanghai, China
yghong@iss.ac.cn

**Yingyi Wu**
University of Chinese Academy of Sciences
Beijing, China
wuyy@ucas.ac.cn

**Guodong Shi**
The University of Sydney
NSW, Australia
guodong.shi@sydney.edu.au

## Abstract

We consider online optimization over Riemannian manifolds, where a learner attempts to minimize a sequence of time-varying loss functions defined on Riemannian manifolds. Though many Euclidean online convex optimization algorithms have been proven useful in a wide range of areas, less attention has been paid to their Riemannian counterparts. In this paper, we study Riemannian online gradient descent (R-OGD) on Hadamard manifolds for both geodesically convex and strongly geodesically convex loss functions, and Riemannian bandit algorithm (R-BAN) on Hadamard homogeneous manifolds for geodesically convex functions. We establish upper bounds on the regrets of the problem with respect to time horizon, manifold curvature, and manifold dimension. We also find a universal lower bound for the achievable regret by constructing an online convex optimization problem on Hadamard manifolds. All the obtained regret bounds match the corresponding results are provided in Euclidean spaces. Finally, some numerical experiments validate our theoretical results.

## 1 Introduction

The *online optimization* has been widely studied in the past decades in online routing, spam filtering, and machine learning [4, 23, 8]. Without a prior knowledge of loss functions, an online convex optimization algorithm predicts solutions before the loss function is revealed.

In this paper, we consider the following Riemannian online convex optimization (R-OCO) problem,

$$\min_{x_t \in \mathcal{K} \subset \mathcal{M}} f_t(x_t), t = 1, 2, \ldots, T, \tag{1}$$

where $\mathcal{M}$ is a complete Riemannian manifold equipped with a Riemannian metric $g$ and $\mathcal{K}$ is a geodesically convex (g-convex) subset of $\mathcal{M}$. Here, $\{f_t\}_{t=1,2,\ldots,T}$ is a sequence of unknown loss functions and every $f_t$ is a geodesically convex (g-convex) function with sufficient smoothness. The R-OCO problem (1) extends the online convex optimization in Euclidean spaces with potential applications in machine learning, such as online principal component analysis (PCA), dictionary learning, and neural networks [28, 19, 25].

---

[*]Correspondence author (Y. Hong, +86-10-82541888 )

35th Conference on Neural Information Processing Systems (NeurIPS 2021).

The R-OCO problem (1) can be understood as a learning process of $T$ rounds. At each round $t = 1, 2, 3, \ldots, T$, an online learner chooses a strategy $x_t$ from the g-convex subset $\mathcal{K}$. Later or simultaneously, the adversary (or nature) produces a g-convex loss function $f_t : \mathcal{K} \to \mathbb{R}$ of which the learner has no prior knowledge. Finally, the learner receives the feedback and suffers the loss $f_t(x_t)$. Generally, there are two types of information feedback. One is the full information feedback, where the entire function $f_t$ is revealed to the learner; other is the bandit feedback, where only the value $f_t(x_t)$ is revealed. The goal of the R-OCO is to minimize the *regret*, defined as

$$R(T) = \sum_{t=1}^{T} f_t(x_t) - \min_{x \in \mathcal{K}} \sum_{t=1}^{T} f_t(x),$$

which measures the difference between the cost by $\{x_t\}_{t=1,\ldots,T}$ and the best-fixed point chosen in hindsight. An algorithm is called no-regret [32], if the regret of the algorithm goes sublinearly with the time horizon $T$.

For carrying out optimization on a manifold, some classical methods treat the manifold as a subset of an ambient Euclidean space and employ Euclidean constrained optimization techniques. For instance, [30] presented an algorithm for the online PCA problem, where the variables were updated in an embedding Euclidean space and then projected onto a manifold. However, in practical applications, the dimension of an embedding Euclidean space can be too high (e.g., the Grassmann manifold [13]), and the projection can be expensive to compute (e.g., the manifold of symmetric positive definite (SPD) matrices [37]). An alternative approach termed *Riemannian optimization* makes use of intrinsic geometry of manifolds so that Riemannian optimization can optimize directly on the manifold as an unconstrained problem, and thus avoiding high dimension embedding and high computing cost for the projection. Furthermore, this viewpoint has shown benefits from the g-convexity, by which a nonconvex optimization problem can be converted into a g-convex one [6]. Consequently, it is important to take a Riemannian approach in our problem (1).

Although there were many existing algorithms for offline manifold optimization problems [2, 31, 5], very few results were obtained about the Riemannian online optimization problem. [35] proposed an online algorithm for estimating hidden Markov chains on Hadamard homogeneous spaces and [9] analyzed Riemannian adaptive methods on products in the regret sense. More recently, [29] studied a zeroth-order online optimization problem on Hadamard manifolds with a sublinear assumption.

**Contribution**   This paper aims to design no-regret algorithms for the R-OCO problem in both full information feedback and bandit feedback. The contribution of this paper is summarized as follows:

- We propose a Riemannian online gradient descent algorithm (R-OGD) for the R-OCO problem in the full information feedback, and then establish upper regret bounds of the R-OGD algorithm for g-convex and strongly g-convex functions.

- We introduce a Riemannian bandit algorithm (R-BAN) for the R-OCO problem in the bandit feedback and then establish an upper regret bound for g-convex functions. Moreover, we develop a key technique to analyze the derivative of a local integration on homogeneous manifolds, which can be applied to estimate gradients in Riemannian optimization and beyond.

- We focus on the worst-case regret and present a universal lower regret bound of R-OCO algorithms with g-convex losses on Hadamard manifolds, which matches the upper bound achieved by the R-OGD algorithm for g-convex functions.

The established lower and upper bounds on the achievable bounds of R-OCO match their counterparts for Euclidean online convex optimization e.g., [38, 24, 20, 1]. We briefly list our results in Table 1.

**Related Work**   The Euclidean online convex optimization was introduced in [38]. Inspired by the gradient descent method, [38] proposed the online gradient descent algorithm (OGD) of which the upper regret bound was proven to be $\mathcal{O}(\sqrt{T})$. Then [24] proceeded with the study of the OGD algorithm and established an upper regret bound $\mathcal{O}(\log T)$ for strongly convex functions. In addition, [1] gave a universal lower bound of $\mathcal{O}(\sqrt{T})$ for online algorithms, which indicated that the bounds in [38] and [24] are essentially optimal. In the bandit setting, [20] provided a detailed exposition of a one-point bandit algorithm. By modifying the gradient in the OGD algorithm to a randomized

Table 1: Comparison of regrets between our work and corresponding results in Euclidean spaces. $T$: the time horizon; $n$: dimension of the manifold; $\zeta$ and $\Lambda$: constants relied on the sectional curvature bound $\kappa$, the dimension $n$ and the domain $\mathcal{K}$.

| | Full information, g-convex | Full information, strongly g-convex | Bandit feedback | Universal lower bound |
|---|---|---|---|---|
| Our Work | $\mathcal{O}\big(\zeta^{\frac{1}{2}}\sqrt{T}\big)$ | $\mathcal{O}\big(\zeta \log T\big)$ | $\mathcal{O}\big(n^{\frac{1}{2}}\zeta^{\frac{1}{4}}\Lambda T^{\frac{3}{4}}\big)$ | $\Omega(\sqrt{T})$ |
| Euclidean | $\mathcal{O}\big(\sqrt{T}\big)$ [38] | $\mathcal{O}\big(\log T\big)$ [24] | $\mathcal{O}\big(n^{\frac{1}{2}}T^{\frac{3}{4}}\big)$[20] | $\Omega(\sqrt{T})$ [1] |

estimator, the upper regret bound attained $\mathcal{O}\big(T^{\frac{3}{4}}\big)$ and $\mathcal{O}\big(T^{\frac{2}{3}}\big)$ for convex loss functions and strongly convex loss functions, respectively. By extending the one-point bandit algorithm, [3] developed a multi-point bandit algorithm and presented upper regret bounds $\mathcal{O}\big(\sqrt{T}\big)$ and $\mathcal{O}\big(\log T\big)$ for convex and strongly convex loss functions. The Riemannian online algorithms proposed in this paper in the full information feedback and the one-point bandit feedback settings are extensions of the algorithms in ([38, 24, 20, 1]) to Riemannian manifolds.

Riemannian optimization has drawn much research attention in the past decades. Many basic algorithms in Euclidean spaces such as the gradient descent method, Newton's method, and trust-region methods have been adapted into a Riemannian setting [2, 5, 31]. Some research of Riemannian stochastic optimization (R-SO) was intended to deal with time-varying optimization problems [12, 36, 37, 35]. Among them, [36] provided the first global complexity analysis for the R-SGD algorithm on geodesically convex problems over Hadamard manifolds, and [35] proposed an online algorithm to deal with hidden Markov chains on Hadamard homogeneous spaces. When performed in batch, R-SO methods are to minimize the average regret in the case of knowing the prior distribution of the loss functions. In these sense, the R-SO can be viewed as a kind of R-OCO problems and R-OCO algorithms can handle settings without prior knowledge.

The results about the R-OCO problem are quite limited. [7] proposed regularized online optimization methods via a Riemann–Lipschitz continuity condition, which focused on convex functions from an ambient Euclidean space. In the full information setting, [9] constructed regret upper bounds of Riemannian adaptive methods for g-convex functions, which required a product manifold structure. When the form of loss function was not available, [29] proposed a zeroth-order online algorithm on Hadamard manifolds for the tracking error in asymptotic sense as well as regret bounds by assuming the sublinearity of the term $V_T$, which is a summation of distance between the minimizer of $f_t$ and $f_{t+1}$. In contrast, the regret bounds established for our online gradient-based/bandit Riemannian optimization algorithms are sublinear for any time, matching those for Euclidean online optimization.

## 2 Preliminaries

**Riemannian manifolds** A *manifold* $\mathcal{M}$ is a topological space locally diffeomorphic to Euclidean spaces. The *tangent space* $T_x\mathcal{M}$ is a linearization of manifold $\mathcal{M}$ at point $x$. A *vector field* $X$ is a map assigning every point $x \in \mathcal{M}$ with a tangent vector $X(x) \in T_x\mathcal{M}$, which can be also viewed as differential operators over smooth functions on $\mathcal{M}$, i.e., the operation $X(f)$ defines a function $X(f)(x) = \lim_{t\to 0} \frac{1}{t}(f(\xi(t)) - f(x))$ on $\mathcal{M}$, where $\xi$ is a curve that starts at $x$ with tangent vector $X(x)$.

A *Riemannian manifold* is a smooth manifold $\mathcal{M}$ equipped with a metric tensor $g$ (or called Riemannian metric), which defines an inner product $\langle \cdot, \cdot \rangle_x$ in every tangent space $T_x\mathcal{M}$ of $x \in \mathcal{M}$. The Riemannian metric $g$ brings a distance structure on $\mathcal{M}$. A curve is a *geodesic* if it locally minimizes the length, which is an analog of a straight line in Euclidean spaces. On Riemannian manifolds, a geodesic is uniquely determined by the start point and initial tangent vector. In this way, the *exponential map* $\exp_x : T_x\mathcal{M} \to \mathcal{M}$ is defined by mapping a vector $X \in T_p\mathcal{M}$ to $\gamma(1) \in \mathcal{M}$ for the geodesic $\gamma$ such that $\gamma(0) = x$ and $\dot{\gamma}(0) = X$.

Curvature reflects the geometry of manifolds. We focus on *sectional curvature*, which is the Gauss curvature of a two-dimensional submanifold. Following [36], we consider the *Hadamard manifold*, which is a simply connected and complete manifold with nonpositive sectional curvature. The Cartan-Hadamard theorem [11] shows that the exponential map $\exp_x(\cdot)$ is a diffeomorphism from the

tangent space $T_x\mathcal{M} \cong R^n$ to the manifold $\mathcal{M}$. Therefore, the exponential map has an global inverse $\exp_x^{-1}(\cdot)$ on Hadamard manifolds, and the distance $d(x,y)$ can be expressed as $\|\exp_x^{-1}(y)\|_x$.

*Isometries* of Riemannian manifolds have been widely studied in differential geometry [11, 10]. An isometry $\phi : \mathcal{M} \to \mathcal{M}$ is a diffeomorphism preserving distance, i.e., $d(x,y) = d(\phi(x), \phi(y))$ for all $x, y \in \mathcal{M}$. It is remarked that all isometries of a Riemannian manifold form a Lie group $G$. A Riemannian manifold is a *homogeneous manifold* if the group of isometries $G$ acts on $\mathcal{M}$ transitively, i.e., for any points $x, y \in \mathcal{M}$ there exists an isometry such that $\phi(x) = y$. Some properties of homogeneous manifolds are used in our work, for example, the Killing field that represents the infinitesimal symmetry of isometries. We leave those properties in the appendix, due to the space limitation.

**Function Classes**  A set $\mathcal{K}$ is called *geodesically convex* (g-convex) if, for any points $x, y \in \mathcal{K}$, there admits a geodesic $\gamma \subset \mathcal{K}$ connecting $x$ and $y$. A function $f : \mathcal{K} \to \mathbb{R}$ is called *geodesically convex* (or g-convex) if for any geodesic $\gamma : [0, 1] \to \mathcal{M}$,

$$f(\gamma(t)) \leq (1-t)f(\gamma(0)) + tf(\gamma(1)), \quad \forall t \in [0, 1].$$

The g-convexity has some equivalent conditions. When $f$ is differentiable, which means that there exists a *gradient* $\nabla f(x)$ such that $\langle \nabla f(x), X \rangle = X(f)(x)$ for any vector field $X$, the g-convexity is equivalent to

$$f(y) \geq f(x) + \langle \nabla f(x), \exp_x^{-1}(y) \rangle, \forall x, y \in \mathcal{M}.$$

Furthermore, if $f$ is twice differentiable, the g-convexity is equivalent to

$$\text{Hess}(f)(x)(X, X) = X(X(f(x)) - (\nabla_X X)f(x) \geq 0, \forall x \in \mathcal{M}, \forall X \in T_x\mathcal{M},$$

where $\nabla$ is the Levi-Civita connection of $\mathcal{M}$, which is an analog of the differential of vector fields in Euclidean spaces (see [17]). A differentiable function $f : \mathcal{M} \to \mathbb{R}$ is *geodesically $\mu$-strongly convex* (or $\mu$-strongly g-convex) if there exists a constant $\mu > 0$ such that for any $x, y \in \mathcal{M}$,

$$f(y) \geq f(x) + \langle \nabla f(x), \exp_x^{-1}(y) \rangle + \frac{\mu}{2}d^2(x, y).$$

We term a function to be *geodesically L-Lipschitz* (or *g-L-Lipschitz*) if there exists a constant $L > 0$ such that,

$$|f(y) - f(x)| \leq L \cdot d(x, y), \forall x, y \in \mathcal{M},$$

When $f$ is differentiable, the g-$L$-Lipschitzness is equivalent to

$$\|\nabla f(x)\| \leq L, \forall x \in \mathcal{M}.$$

## 3  Riemannian Online Convex Optimization with Full Information Feedback

This section is devoted to the study of the R-OCO problem in the full information feedback. Here, we first propose our R-OGD algorithm and then analyze upper regret bounds of R-OGD for both g-convex and strongly g-convex functions. In addition, a universal lower regret bound is presented to illustrate that the regret bound of the R-OGD algorithm is tight up to a constant in the g-convex case.

### 3.1  Riemannian Online Gradient Algorithm

In the full information setting, we consider the following assumptions, which were used in the literature of Euclidean online convex optimization and Riemannian optimization (e.g., [38, 36, 5]).

**Assumption 1.** *There exists $x^\star \in \mathcal{M}$ such that $x^\star = \arg\min \sum_{t=1}^T f_t(x)$.*

**Assumption 2.** *$(\mathcal{M}, g)$ is a Hadamard manifold with the sectional curvature lower bounded by a constant $-\kappa$ ($\kappa \geq 0$).*

**Remark 1.** The Hadamard manifold plays an important role in Riemannian geometry [22]. Some well-known spaces, such as the Euclidean space $\mathbb{R}^n$, the hyperbolic space $H^n$, and the manifold of SPD matrices, are all Hadamard manifolds [36, 5].

**Assumption 3.** *The g-convex set $\mathcal{K}$ is a bounded set with diameter $D$, i.e.,*

$$d(x, y) \leq D, \forall x, y \in \mathcal{K}.$$

**Assumption 4.** *For all $t = 1, \ldots, T$, $f_t$ are differentiable and g-L-Lipschitz.*

We now propose our Riemannian online gradient descent algorithm (R-OGD) in Algorithm 1, where the exponential map replaces the vector addition in the Euclidean online gradient descent [38].

---

**Algorithm 1:** Riemannian Online Gradient Descent Algorithm (R-OGD)

---

**Input:** Manifold $\mathcal{M}$, time $T$, step sizes $\{\alpha_t\}$
**Output:** $\{x_t\}_{t=1,\ldots,T}$
**for** $t = 1$ *to* $T$ **do**
    Play $x_t$ and observe the function $f_t$;
    Update
$$x_{t+1} = \mathcal{P}_{\mathcal{K}}(\exp_{x_t}(-\alpha_t \nabla f_t(x_t))),$$
    where $\mathcal{P}_{\mathcal{K}}$ is the Riemannian projection mapping of $x$ onto $\mathcal{K}$, that is,
    $\mathcal{P}_{\mathcal{K}}(x) := \arg\min_{y \in \mathcal{K}} d(x, y)$;
    Return $x_{t+1}$, and suffer from the loss $f_t(x_t)$;
**end**

---

## 3.2 Regret Upper Bounds

In Theorems 1 and 2 we present upper bounds of the regret along the R-OGD algorithm for g-convex and strongly g-convex functions, respectively. Take $\zeta(\kappa, d) = \frac{\sqrt{\kappa} \cdot d}{\tanh\left(\sqrt{\kappa} \cdot d\right)}$. By direct observation, $\zeta$ is an increasing function of the variables $\kappa$ and $d$ when $\kappa d \geq 0$.

**Theorem 1** (Convex Case). *Suppose that Assumptions 1-4 hold, and $f_t$ is g-convex for any $t = 1, \ldots, T$. Then the R-OGD algorithm with step sizes $\{\alpha_t = \frac{D}{L\sqrt{\zeta(\kappa, D)t}}\}$ guarantees the following regret bound for all $T \geq 1$,*
$$R(T) \leq \frac{3}{2} DL\sqrt{\zeta(\kappa, D)T}.$$

**Theorem 2** (Strongly-convex Case). *Suppose that Assumptions 1-4 hold, and $f_t$ is $\mu$-strongly g-convex for any $t = 1, \ldots, T$. Then the R-OGD algorithm with step sizes $\{\alpha_t = \frac{1}{\mu t}\}$ guarantees the following regret bound for all $T \geq 1$,*
$$R(T) \leq \frac{L^2 \zeta(\kappa, D)}{2\mu}(1 + \log T).$$

The proofs of Theorems 1 and 2 are in the appendix. A major challenge in proving Theorems 1 and 2 is that there is no vector space structure on Riemannian manifolds. Thanks to the trigonometric distance bound proposed in [36], we manage to obtain the regret $\mathcal{O}(\sqrt{T})$ and $\mathcal{O}(\log T)$ for g-convex and strongly g-convex loss functions, respectively. By gradually moving $\kappa$ to zero, the results recover the regret bounds for Euclidean gradient descent in [38] and [24].

Theorems 1 and 2 also reveal the influence of curvature on the regret bounds. Since $\zeta(\kappa, d)$ is an increasing function of $\kappa$, the upper regret bounds in the R-OGD algorithm are larger than those in Euclidean spaces and the increase of $\kappa$ raises the upper regret bound. Therefore, a proper Riemannian metric should be chosen in the optimization to avert the high sectional curvature bound.

## 3.3 Regret Lower Bound

This section is intended to answer the question of whether there exists an algorithm that attains a tighter regret bound than $\mathcal{O}(\sqrt{T})$ for g-convex functions. Theorem 3 provides a negative answer.

**Theorem 3.** *Suppose that Assumptions 1-4 hold. Then for any Hadamard manifold $\mathcal{M}$, the Riemannian online convex optimization incurs the regret $\Omega(DL\sqrt{T})$ for any possible strategy in the worst case.*

The proof of Theorem 3 is in the appendix. The result illustrates that, as in Euclidean spaces, the regret of a Riemannian online comvex algorithm can not be less than $\Omega(\sqrt{T})$ in the worst case. Moreover, Theorem 3 shows that the regret of the R-OGD algorithm in Theorem 1 is tight up to a constant.

# 4 Riemannian Online Convex Optimization with Bandit Feedback

In this section, we consider the Riemannian online convex optimization with the one-point bandit feedback. We first present the Riemannian bandit algorithm (R-BAN) on Hadamard homogeneous manifolds and then analyze the (expected) upper regret bound for our algorithm. In the rest of this section, for any point $x$, we denote $B_\delta(x)$ as the geodesic ball centered at $x$ with radius $\delta$, and $S_\delta(x)$ as the geodesic sphere centered at $x$ with radius $\delta$.

## 4.1 Riemannian Bandit Algorithm

In the bandit setting, Assumptions 2-4 are slightly modified as follows.

**Assumption 5.** *$\mathcal{M}$ is an n-dimensional homogeneous Hadamard manifold with the sectional curvature lower bounded by a constant $-\kappa$ ($\kappa \geq 0$).*

**Remark 2.** The homogeneous Hadamard manifold has been widely studied in differential geometry [10, 11]. The property of homogeneity has received much attention in machine learning [33, 35, 14]. It has been seen that many manifolds often considered in Riemannian optimization, such as the Euclidean space $\mathbb{R}^n$, the Hyperbolic space $H^n$, and the manifold of SPD matrices, are Hadamard homogeneous manifolds.

Note that on Hadamard homogeneous manifolds, the volume and surface area of a geodesic ball is only related to the radius but not to the center of the ball, thus we denote $V_\delta$ as the volume of $B_\delta(x)$ and $S_\delta$ as the surface area $((n-1)$-dim volume) of $S_\delta(x)$ for all $x$ in the manifold $\mathcal{M}$.

**Assumption 6.** *There exists an interior point $p \in \mathcal{K}$ such that the set $\mathcal{K}$ contains a ball with radius $r$ centered at $p$, and is also contained in a ball with radius $D$, i.e.,*

$$B_r(p) \subset \mathcal{K} \subset B_D(p).$$

**Assumption 7.** *For any $t = 1, \ldots, T$, $f_t$ is differentiable, g-L-Lipschitz and bounded by $C$.*

Inspired by the Euclidean bandit algorithm, we replace the gradient $\nabla f_t(x_t)$ with a randomized estimator $g_t$ and propose our R-BAN in Algorithm 2 on Hadamard homogeneous manifolds.

---

**Algorithm 2:** Riemannian Bandit Algorithm (R-BAN)

---

**Input:** Manifold $\mathcal{M}$, time $T$, step size $\alpha$, parameters $\delta, \tau$.
**Output:** Sequence $\{x_t\}_{t=1,\ldots,T}$
**for** $t = 1$ *to* $T$ **do**
    Pick $x_t$ uniformly from $S_\delta(y_t)$;
    Play $x_t$ and observe $f_t(x_t)$;
    Construct the gradient estimator

$$g_t = f_t(x_t) \frac{\exp_{y_t}^{-1}(x_t)}{\|\exp_{y_t}^{-1}(x_t)\|};$$

    Update $y_t$ with the rule

$$y_{t+1} = P_{(1-\tau)\mathcal{K}}(\exp_{y_t}(-\alpha g_t)),$$

    where we denote $P_{(1-\tau)K}$ as the projection mapping onto the shrinking set
    $(1-\tau)\mathcal{K} = \{\exp_p((1-\alpha)u)|u = \exp_p^{-1}(x), x \in \mathcal{K}\}$.
    Return $x_t$ and suffer from the loss $f_t(x_t)$;
**end**

---

## 4.2 Challenges from Geometry

Since Algorithm 2 is an extension of the Euclidean bandit algorithm in [20], it is worth reviewing the analysis in [20]. In the Euclidean setting, we uniformly choose $x_t$ on the $S_\delta(y_t)$ and update $y_t$ by the rule

$$\begin{cases} \tilde{g}_t = f(x_t) \frac{x_t - y_t}{\|x_t - y_t\|}, \\ y_{t+1} = \mathcal{P}_{(1-\tau)\mathcal{K}}(y_t - \alpha \tilde{g}_t). \end{cases} \tag{2}$$

The basic idea for the analysis is to introduce the *smoothed loss function* [20]

$$\hat{f}_t^E(x) = \mathbb{E}_{u \in B_\delta(x)}[f_t(u)] = \frac{1}{V_\delta} \int_{B_\delta(x)} f_t(u) du,$$

where $\hat{f}_t^E$ is a convex approximation of $f_t$ when $\delta$ is small. It is shown that $\frac{n}{\delta} \tilde{g}_t$ is an unbiased estimator of the gradient $\nabla \hat{f}_t^E(y_t)$, hence the bandit algorithm is actually an expected gradient descent method [20] with the loss function $\hat{f}_t^E$. In this way, an Euclidean regret bound of the bandit algorithm is established by [20].

Back to the Riemannian case, we attempt to generalize the analysis of [20] in parallel by defining the "Riemannian version" of the smoothed loss function,

$$\hat{f}_t(x) = \mathbb{E}_{u \in B_\delta(x)}[f_t(u)] = \frac{1}{V_\delta} \int_{B_\delta(x)} f_t(u)\omega,$$

where $\omega$ is the volume element with respect to the metric $g$. Analyzing this smoothed loss function in the Riemannian space, however is fundamentally challenging due to the following two reasons.

**(i) The gradient is hard to compute.** Computing the gradient of $\hat{f}_t$ is quite different from that in Euclidean spaces, due to the absence of the commutativity of the derivative operator $\nabla$ and the integration operator $\int_{B_\delta(y_t)}$. In Euclidean spaces, the derivative operator $\nabla$ commutes with the integration operator $\int_{B_\delta(y_t)}$. Accordingly, for the Euclidean smoothed loss function $\hat{f}_t^E$,

$$\nabla \hat{f}_t^E(y_t) = \frac{1}{V_\delta} \nabla \int_{B_\delta(y_t)} f_t(u) du = \frac{1}{V_\delta} \int_{B_\delta(y_t)} \nabla f_t(u) du, \tag{3}$$

which implies $\frac{n}{\delta} E[\tilde{g}_t] = \nabla \hat{f}_t^E(y_t)$. However, on Riemannian manifolds the derivative operator $\nabla$ does not commute with the integration operator $\int_{B_\delta(y_t)}$. Consequently, the equation (3) fails for functions on Riemannian manifolds.

**(ii) The convexity may be lost.** Another essential challenge for regret analysis is the convexity of $\hat{f}_t$. In Euclidean spaces, one can easily conclude the convexity of $\hat{f}_t^E$. However, the convexity may not hold for a Riemannian manifold. Through calculation, the Hessian of $\hat{f}_t$ on Riemannian manifolds is

$$\text{Hess}(\hat{f}_t)(X, X) = \frac{1}{V_\delta} \int_{B_\delta(x)} (\text{Hess}(f_t)(\eta, \eta)(u) + \langle \nabla_\eta \eta, \nabla f_t(u) \rangle)\omega,$$

where $\eta$ is a Killing field such that $\eta(x) = X$. Since the quadratic form $\text{Hess}(f_t)(\eta, \eta)(x) + \langle \nabla_\eta \eta, \nabla f_t(x) \rangle$ can be negative at some $\eta \in T_p\mathcal{M}$, the g-convexity of $\hat{f}_t$ is violated for some small $\delta$.

### 4.3 Gradient Bound and Approximate g-Convexity

To address the difficulty (i), we propose a key technique to analyze the derivative of local integration by introducing the Killing vector field. With the help of this technique, we manage to compute the gradient of $\hat{f}_t$ in Lemma 1.

**Lemma 1.** *Let $\mathcal{M}$ be a homogeneous Hadamard manifold, and $f$ be a $C^1$ function on $\mathcal{M}$. Then the smoothed function $\hat{f}(x) = \frac{1}{V_\delta} \int_{B_\delta(x)} f\omega$ satisfies,*

*1) $\nabla \hat{f}(x) = \frac{1}{V_\delta} \int_{S_\delta(x)} f(u) \frac{\exp_x^{-1}(u)}{\|\exp_x^{-1}(u)\|} = \frac{S_\delta}{V_\delta} \mathbb{E}_{u \in S_\delta(x)} \left[ f(u) \frac{\exp_x^{-1}(u)}{\|\exp_x^{-1}(u)\|} \right], \forall x \in \mathcal{M};$*

*2) If $|f(x)| < C$, then*

$$\|\nabla \hat{f}(x)\| \leq \frac{S_\delta}{V_\delta} C \leq \frac{n}{\delta} C + n\kappa\delta C$$

*for all $\delta > 0$.*

**Remark 3.** The proof of Lemma 1 can be found in the appendix. The first part of the lemma establishes a gradient estimator of $\hat{f}_t(x)$, while the second part gives us an easy-to-compute bound of the gradient. In the proof, we develop a key technique that transforms a derivation of integration on $B_\delta(x)$ to a integration of derivative of corresponding Killing vector field on $B_\delta(x)$, i.e.,

$$X\Big(\int_{B_\delta(x)} f(u)\omega\Big) = \int_{B_\delta(x)} \eta(f(u))\omega, \tag{4}$$

where $\eta$ is a Killing field with $\eta(x) = X$. This technique does not rely on the curvature and other specific manifold structures. Hence, the technique can be a basic tool for optimization on homogeneous spaces and maybe in broader areas.

For difficulty (ii), we notice that though the function $\hat{f}_t$ may not be g-convex, it is very close to be g-convex.

**Lemma 2.** *Suppose that $\mathcal{M}$ is a Hadamard homogeneous manifold, and $\mathcal{K}$ is a convex and bounded set of $\mathcal{M}$. Then there exists a constant $\rho \geq 0$ only related to the set $\mathcal{K}$ such that for any g-convex and g-L-Lipschitz function $f$,*

$$\hat{f}(y) - \hat{f}(x) - \langle \nabla \hat{f}(x), \exp_x^{-1}(y)\rangle \geq -2\rho\delta L, \forall x, y \in \mathcal{K}.$$

The proof of Lemma 2 is in the appendix. It is worth mentioning that the constant $\rho$ describes how close a smoothed function is to be g-convex. Notice that,

$$\rho = \sup_{x,y,u\in\mathcal{K}} |\frac{1}{\sqrt{G}}\frac{\partial}{\partial x_i}\big(\sqrt{G}\exp_u^{-1}\phi(u)\big)^i| \quad s.t. \quad \phi(x) = y$$

does not depend on the function $\hat{f}_t$, and the time $T$. Moreover, for a certain manifold $\mathcal{M}$, once the set $\mathcal{K}$ is fixed and the explicit expression of $\phi$ is given, we can compute the constant $\rho$ as a finite number. We briefly list the bound of $\phi$ in the following two kind of manifolds.

(1) Let manifold $\mathcal{M}$ be a Euclidean space, then we can find the isometry $\phi(z) = z + y - x$. Hence we can conclude that $\rho = 0$ and $\hat{f}$ is convex, which coincides with the result in Euclidean spaces.

(2) Let $\mathcal{M}$ be a 2-dimensional Poincaré disk, and then the isometry $\phi$ from $x$ to $y$ has the closed form of

$$\phi = \phi_x \circ \phi_y,$$

where $\phi_x(z) = \frac{x-z}{1-\bar{x}z}$ and $\phi_y(z) = \frac{y-z}{1-\bar{y}z}$. Therefore, if $\mathcal{K}$ has diameter $D$, we can figure out a bound of $\rho$ in

$$\rho \leq 16\frac{1+\tanh(D/2)}{1-\tanh(D/2)}\Big(\frac{1}{1-\tanh(2D)^2} + \frac{D}{\tanh(D/2)}\Big),$$

which implies that $\rho$ may grow exponentially with respect to $D$.

Although the value of $\rho$ is generally difficult to calculate, our algorithm analysis and parameter selection do not depend on the specific value of $\rho$ (see Theorem 4).

## 4.4 Regret Bound

With the above effort, we carry out the analysis of the expected regret bounds of Algorithm 2. Denote $B = n\kappa$, $\Delta = BCD\sqrt{\zeta(\kappa, D)} + 3L + 2C/r$ and $\Lambda = \sqrt{\Delta} + \frac{2\rho L}{\sqrt{\Delta}}$.

**Theorem 4.** *Suppose that Assumptions 1 and 5-7 hold, and $f_t$ is g-convex for any $t = 1, \dots, T$. Take $\tau = \frac{\delta}{r}$, $\alpha = \frac{D}{C\sqrt{\zeta(\kappa,D)T}}$, $\delta = T^{-\frac{1}{4}}\sqrt{\frac{CDn\sqrt{\zeta(\kappa,D)}}{\Delta}}$. Then the expected regret of Algorithm 2 is upper bounded by*

$$\mathbb{E}[R(T)] \leq 2T^{\frac{3}{4}}\sqrt{nCD\sqrt{\zeta(\kappa, D)}}\Lambda.$$

The proof of Theorem 4 can be seen in the appendix. Theorem 4 shows that the regret of the Riemannian bandit algorithm achieves $\mathcal{O}\left(T^{\frac{3}{4}}\right)$ for g-convex loss functions on homogeneous Hadamard manifolds, which is the same as the regret bounds in Euclidean spaces [20].

Our work is different from the result in [29] in two aspects. First, in [29] the sublinear regrets bounds depends on the assumption of sublinearity of the term $V_T$, while in Theorem 4, the sublinear regret bound is expressed explicitly with the time horizon $T$ with no additional assumptions. Second, to estimate the gradient, [29] used a biased estimator of the gradient, while in our bandit algorithm the gradient estimator $g_t$ of the smoothed function is unbiased.

## 5 Numerical Experiment

We validate our findings on Riemannian manifolds in both g-convex and strongly g-convex losses. We also compare our results with the Riemannian zeroth online (R-OZO) algorithm [29] if possible (the R-OZO is only suitable for strongly g-convex cases). All experiments are performed with the help of Pymanopt toolbox [34][2] on a 64 bit Windows platform with a 3.4 GHz CPU (AMD Ryzen5 2600) and the code used for the numerical experiments is provided in the supplementary materials.

### 5.1 Strongly Convex Cases

For strongly g-convex cases, we apply our R-OGD and R-BAN algorithms to the Fréchet mean problem, which is also known as finding the Riemannian centroid of a set of points on a manifold with many applications, such as diffusion tensor magnetic resonance imaging (DT-MRI), radar signal processing, and batch normalization of neural networks [16, 27, 15]. In this subsection, we study an online form of the Fréchet mean problem to average a set of $N$ time-varying points $\{A_{t,1}, A_{t,2}, A_{t,3}, \ldots, A_{t,N}\}$ on a manifold. The loss function is defined as

$$f_t(X_t) = \frac{1}{N} \sum_{i=1}^{N} d^2(X_t, A_{t,i}), t = 1, 2, \ldots, T,$$

where $d(X, Y)$ is the Riemannian distance of the manifold. It is remarked that $f_t$ is g-convex and 2-strongly-convex [18] so that we can apply our algorithms. In particular, we test the R-OGD and the R-BAN algorithm, respectively, on the manifold of SPD matrices and in the hyperbolic space, and we also compare our R-BAN algorithm with the R-OZO in [29].

**Fréchet mean on the SPD Manifold**   On the manifold of SPD matrices $\{X \in \mathbb{R}^{n \times n} | X^T = X, X \succ 0\}$, we run the R-OGD algorithm with two kinds of step size: the step size for convex case $\alpha_t = D/(L\sqrt{\zeta(\kappa, D)t})$ (R-OGD-C) and the step size for strongly g-convex case $\alpha_t = 1/(2t)$ (R-OGD-SC). In these two cases, we set $[n, N, T] = [100, 10, 1000]$. Matrices $A_{i,t}$ are randomly generated by the method in Pymanopt toolbox. We plot the average regret $R(t)/t$ versus the learning round $t$ and the running time in Figures 1(a)-1(b). As seen, the regrets of the R-OGD algorithm go sublinearly with $t$ and the average regret of the R-OGD-SC converges faster than that of R-OGD-C in terms of the iteration as well as the running time, which are consistent with the results in Theorems 1 and 2.

**Fréchet mean on the Hyperbolic Space**   We test the performance of the R-BAN algorithm in the hyperbolic space $H^n = \{x \in \mathbb{R}^{n+1} | -x_{n+1}^2 + \sum_{i=1}^{n} x_i^2 = -1\}$. The data $A_{i,t}$ is randomly generated by normal Gaussian distributions. Consider the case $[n, N, T] = [100, 10, 10000]$ and set $\delta = 0.399$ (which is four times the theoretical value) and $\alpha = 0.006$. Figures 1(b) and 2(b) compare the average performance between the R-BAN algorithm and the R-OZO algorithm for 100 random runs. We observe that our R-BAN algorithm can achieve a sublinear expected regret with less information, since the R-OZO needs function values of two points in this case.

### 5.2 Convex Cases

An example of Riemannian optimization problems, which is g-convex but not strongly g-convex, is the operator scaling problem defined on the manifold of SPD matrices $\{X \in \mathbb{R}^{n \times n} | X^T = X, X \succ 0\}$.

---

[2]https://www.pymanopt.org/, BSD 3-Clause License

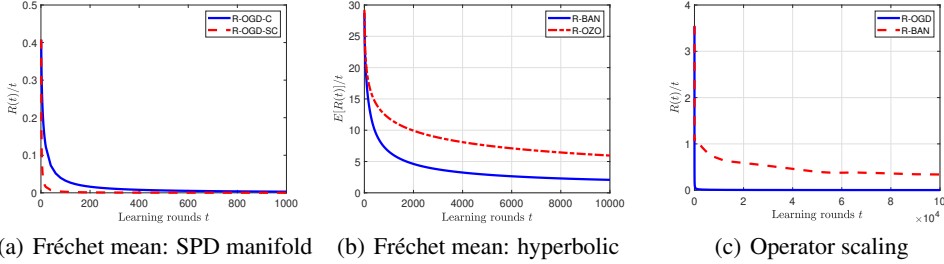

(a) Fréchet mean: SPD manifold     (b) Fréchet mean: hyperbolic     (c) Operator scaling

Figure 1: Average regret vs learning rounds

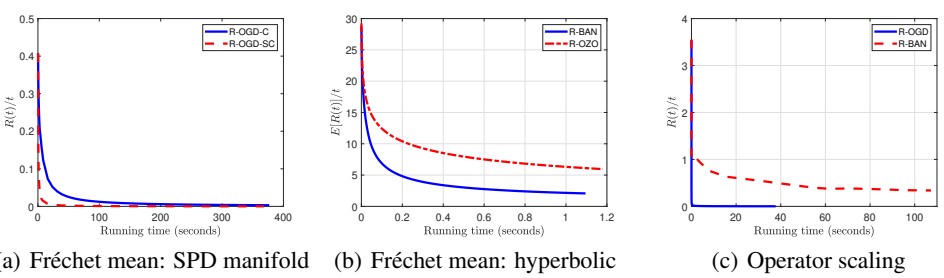

(a) Fréchet mean: SPD manifold     (b) Fréchet mean: hyperbolic     (c) Operator scaling

Figure 2: Average regret vs running time

The operator scaling problem has drawn abundant interest in many areas, such as computing non-commutative rank [26] and computing Brascamp-Lieb constants [21]. In this subsection, we study an online form of the operator scaling problem. Given a tuple of time-varying matrices $(A_{t,1}, \ldots, A_{t,N})$, the online operator scaling can be formulated in terms of minimizing the log capacity of operator $T_t(X) = \sum_{i=1}^{N} A_{t,i} X A_{t,i}^T$, that is

$$f_t(X_t) = \log \det(T(X_t)) - \log \det(X_t), t = 1, 2, \ldots, T.$$

We test our R-OGD and R-BAN algorithms for the case $[n, N, T] = [5, 2, 100000]$. The entries of $A_{i,t}$ are generated from the normal Gaussian distribution. We test the R-OGD with with taking the Lipschitz constant $L = 2$ and test the R-BAN with $\delta = 0.22$ (which is five times the theoretical value) and $\alpha = 0.002$ for 100 different runs. The result in Figures 1(c) and 2(c) again shows the (expected) sublinear regret in this case, which supports our theoretical results.

## 6 Conclusion

We considered an online optimization problem on Riemannian manifolds in the full information and bandit feedback setting. We developed the R-OGD algorithm on Hadamard manifolds and the R-BAN algorithm on Hadamard homogeneous manifolds. The upper regret bounds of the R-OGD and R-BAN algorithm, together with a universal lower regret bound were established with the influence of curvature clearly indicated. All of the regret bounds matched their Euclidean counterpart.

A limitation of our work is that we do not take retraction into consideration. A retraction map is a cheap approximation of the exponential map on manifolds and is a sensible choice in many real scenarios. In future work, we intend to design Riemannian online optimization methods with the retraction map, so that the resulting algorithms can be more effective in large-scale optimization problems.

## Acknowledgement

This work is supported in part by Shanghai Municipal Science and Technology Major Project under Grant 2021SHZDZX0100 and the National Natural Science Foundation of China under Grant 61733018, and in part by Australian Research Council under Grant DP190103615.

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
