# A Proofs of Theorems 1 and 2

We first recall the following lemma in [8], which shows a relationship between iteration points $x_t$ and $x_{t+1}$.

**Lemma 3** ([8]). *Suppose that $\mathcal{M}$ is a Hadamard manifold with the sectional curvature lower bounded by $-\kappa$ ($\kappa > 0$) and $\mathcal{K} \subset \mathcal{M}$ is a g-convex set. Then, for any $x, x_t \in \mathcal{K}$, $x_{t+1} = \mathcal{P}_{\mathcal{K}}(\exp_{x_t}(-\alpha_t g_t))$ satisfies*

$$\langle -g_t, \exp_{x_t}^{-1}(x) \rangle \le \frac{1}{2\alpha_t}(d^2(x_t, x) - d^2(x_{t+1}, x)) + \frac{1}{2}\zeta(\kappa, d(x_t, x))\alpha_t \|g_t\|^2.$$

Then we begin our proofs of Theorems 1 and 2.

*Proof of Theorem 1.* By the g-convexity, we have

$$f_t(x_t) - f_t(x^*) \le \langle -\nabla f_t(x_t), \exp_{x_t}^{-1}(x^*) \rangle.$$

Recalling Lemma 3 gives

$$f_t(x_t) - f_t(x^*) \le \frac{1}{2\alpha_t}(d^2(x_t, x^*) - d^2(x_{t+1}, x^*)) + \frac{1}{2}\zeta(\kappa, d(x_t, x^*))\alpha_t \|\nabla f_t(x_t)\|^2.$$

With the Lipschitz constant $L$, we have

$$f_t(x_t) - f_t(x^*) \le \frac{1}{2\alpha_t}(d^2(x_t, x^*) - d^2(x_{t+1}, x^*)) + \frac{1}{2}\zeta(\kappa, d(x_t, x^*))L^2\alpha_t. \qquad (5)$$

Summing (5) from 1 to $T$, we obtain

$$\begin{aligned}
R(T) &= \sum_{t=1}^{T} f_t(x_t) - \sum_{t=1}^{T} f_t(x^\star) \\
&\le \sum_{t=1}^{T} \frac{1}{2\alpha_t}\big(d^2(x_t, x^\star) - d^2(x_{t+1}, x^\star)\big) + \sum_{t=1}^{T} \frac{1}{2}\zeta\big(\kappa, d(x_t, x^*)\big)L^2\alpha_t \\
&= \sum_{t=2}^{T} d^2(x_t, x^\star)\big(\frac{1}{2\alpha_t} - \frac{1}{2\alpha_{t-1}}\big) + \frac{1}{2}\alpha_1 d^2(x_1, x^\star) + \frac{1}{2}L^2 \sum_{t=1}^{T} \zeta\big(\kappa, d(x_t, x^*)\big)\alpha_t.
\end{aligned}$$

Since the set $\mathcal{K}$ has diameter $D$, $d(x_t, x^*) \le D$ and $\zeta\big(\kappa, d(x_t, x^*)\big) \le \zeta(\kappa, D)$ for every $t = 1, 2, \dots, T$, which implies

$$\begin{aligned}
R(T) &\le D^2 \sum_{t=2}^{T} (\frac{1}{2\alpha_t} - \frac{1}{2\alpha_{t-1}}) + D^2 \frac{1}{2}\alpha_1 + \frac{1}{2}\zeta(\kappa, D)L^2 \sum_{t=1}^{T} \alpha_t \\
&= D^2 \frac{1}{2\alpha_T} + \frac{1}{2}\zeta(\kappa, D)L^2 \sum_{t=1}^{T} \alpha_t,
\end{aligned}$$

Setting $\alpha_t = \frac{D}{L\sqrt{\zeta(\kappa, D)t}}$, we get

$$\begin{aligned}
R(T) &\le \frac{DL\sqrt{\zeta(\kappa, D)}}{2}\sqrt{T} + \frac{1}{2}\zeta(\kappa, D)L^2 \sum_{t=1}^{T} \alpha_t \\
&\le \frac{DL\sqrt{\zeta(\kappa, D)}}{2}\sqrt{T} + \frac{1}{2}\zeta(\kappa, D)L^2 \frac{2D}{L\sqrt{\zeta(\kappa, D)}}\sqrt{T} \\
&= \frac{3}{2}DL\sqrt{\zeta(\kappa, D)T},
\end{aligned}$$

The second inequality is based on the inequality $\sum_{t=1}^{T} \frac{1}{\sqrt{t}} \le 2\sqrt{T}$, and then we complete our proof. $\qquad \square$

*Proof of Theorem 2.* By the strong g-convexity, we have

$$f_t(x_t) - f_t(x^*) \leq \langle -\nabla f_t(x_t), \exp_{x_t}^{-1}(x^*) \rangle - \frac{\mu}{2} d^2(x_t, x^*).$$

With the help of Lemma 3 and the Lipschitz constant $L$, we have

$$f_t(x_t) - f_t(x^*) \leq \frac{1}{2\alpha_t} \left( d^2(x_t, x^*) - d^2(x_{t+1}, x^*) \right) + \frac{1}{2} \zeta\left(\kappa, d(x_t, x^*)\right) L^2 \alpha_t - \frac{\mu}{2} d^2(x_t, x^*).$$
(6)

Summing (6) from 1 to $T$, we obtain

$$R(T) = \sum_{t=1}^{T} f_t(x_t) - \sum_{t=1}^{T} f_t(x^\star)$$

$$\leq \sum_{t=1}^{T} \frac{1}{2\alpha_t} \left( d^2(x_t, x^\star) - d^2(x_{t+1}, x^\star) \right) + \sum_{t=1}^{T} \frac{1}{2} \zeta(\kappa, d(x_t, x^*)) L^2 \alpha_t - \sum_{t=1}^{T} \frac{\mu}{2} d^2(x_t, x^*)$$

$$= \sum_{t=2}^{T} d^2(x_t, x^\star)\left(\frac{1}{2\alpha_t} - \frac{1}{2\alpha_{t-1}} - \frac{\mu}{2}\right) + d^2(x_1, x^*)\left(\frac{\alpha_1}{2} - \frac{\mu}{2}\right) + \frac{1}{2} L^2 \sum_{t=1}^{T} \zeta(\kappa, d(x_t, x^*)) \alpha_t.$$

Substituting $d(x_t, x^*) \leq D$ and $\zeta(\kappa, d(x_t, x^*)) \leq \zeta(\kappa, D)$ for $t = 1, 2, \dots, T$, we obtain

$$R(T) \leq \sum_{t=2}^{T} D^2\left(\frac{1}{\alpha_t} - \frac{1}{2\alpha_{t-1}} - \frac{\mu}{2}\right) + D^2\left(\frac{1}{2\alpha_1} - \frac{\mu}{2}\right) + \frac{1}{2} L^2 \sum_{t=1}^{T} \zeta(\kappa, d(x_t, x^*)) \alpha_t$$

$$= D^2\left(\frac{1}{2\alpha_T} - \frac{\mu T}{2}\right) + \frac{1}{2} \zeta(\kappa, D) L^2 \sum_{t=1}^{T} \alpha_t.$$

Setting $\alpha_t = \frac{1}{\mu t}$, we get

$$R(T) \leq 0 + \frac{1}{2} \zeta(\kappa, D) L^2 \sum_{t=1}^{T} \alpha_t \leq \frac{\zeta(\kappa, D) L^2}{2\mu} (1 + \log T).$$

The second inequality follows from the inequality $\sum_{t=1}^{T} \frac{1}{t} \leq 1 + \log T$, and then we complete our proof. $\square$

# B    Proof of Theorem 3

In this appendix, we first introduce an instance of Riemannian online convex optimization called Riemannian online Busemann optimization (ROBO) and then prove Theorem 3 by analyzing the worst-case regret of the ROBO problem.

## B.1    Riemannian Online Busemann Optimization

It is shown that the Busemann function [1] is useful to study the large-scale geometry of Hadamard manifolds.

**Definition 1** ([1]). Let $\mathcal{M}$ be a Hadamard manifold and $\gamma : [0, \infty)$ be a geodesic ray on $\mathcal{M}$ with $\|\dot{\gamma}(0)\| = 1$. Then the Busemann function with $\gamma$ is defined as

$$f_\gamma(x) = \lim_{t \to \infty} \left( d(x, \gamma(t)) - t \right).$$

Here are some properties of the Busemann function.

**Lemma 4** ([1]). *For a Busemann function $f_\gamma$,*

   *1) $f_\gamma$ is g-convex;*

2) $\nabla f_\gamma(\gamma(t)) = \dot\gamma(t)$ *for every* $t \in [0, \infty)$;

3) $\|\nabla f_\gamma(x)\| \le 1$ *for every* $x \in \mathcal{M}$.

Next, we introduce some notations. Let $D, L > 0$ be two constants, $\mathcal{M}$ be a Hadamard manifold, $p \in \mathcal{M}$ and $\gamma : \mathbb{R} \to \mathcal{M}$ be a geodesic with conditions $\|\dot\gamma(0)\| = 1$ and $\gamma(0) = p$. Then we consider an instance of R-OCO problem termed *Riemannian online Busemann optimization* (ROBO) on $\mathcal{M}$, where the g-convex set $\mathcal{K}$ is the ball centered $p$ with radius $D$ ,i.e.,

$$\mathcal{K} = \{x \in \mathcal{M} | d(x, p) \le D\},$$

and the loss function $f_t$ is randomly and uniformly chosen from the set

$$\{Lf_+, Lf_-\},$$

where $f_+$ and $f_-$ are Busemann functions related to the geodesic rays $\gamma_+(t) = \gamma(t)$ and $\gamma_-(t) = \gamma(-t)$. The regret of the ROBO problem is

$$R(T) = \sum_{t=1}^{T} f_t(x_t) - \min_{x \in \mathcal{K}} \sum_{t=1}^{T} f_t(x).$$

In the last part of the subsection, we propose a lemma about the minimum of $\sum_{t=1}^{T} f_t(x)$.

**Lemma 5.** *The minimum of* $af_+(t) + bf_-(t), (a, b \in \mathbb{N})$ *in* $\mathcal{K}$ *is* $-|a - b|D$.

*Proof.* By the g-convexity of $f_+$ and $f_-$, we have

$$af_+(x) + bf_-(x) \ge af_+(p) + bf_-(p) + \langle a\nabla f_+(p) + b\nabla f_-(p), \exp_p^{-1}(x)\rangle, \forall x \in \mathcal{K}.$$

Because $\nabla f_+(p) = \dot\gamma(0)$, $\nabla f_-(p) = -\dot\gamma(0)$ and $f_\pm(p) = 0$, we have

$$af_+(x) + bf_-(x) \ge \langle (a - b)\dot\gamma(0), \exp_p^{-1}(x)\rangle, \forall x \in \mathcal{K}.$$

Moreover, since $\|\dot\gamma(0)\| = 1$ and $\|\exp_p^{-1}(x)\| = d(x, p) \le D$, we have

$$\min_{x \in \mathcal{K}} af_+(x) + bf_-(x) \ge \min_{x \in \mathcal{K}} \langle (a - b)\dot\gamma(0), \exp_p^{-1}(x)\rangle \ge -|a - b|D. \tag{7}$$

However, we see that $af_+(\gamma(D)) + bf_-(\gamma(D)) = (b - a)D$ and $af_+(\gamma(-D)) + bf_-(\gamma(-D)) = (a - b)D$, which imply

$$\min_{x \in \mathcal{K}} af_+(x) + bf_-(x) \le \min\{(b - a)D, (a - b)D\} = -|a - b|D. \tag{8}$$

Following from (7) and (8), we complete our proof. $\square$

### B.2 Proof of Theorem 3

We begin our proof with an analysis of the worst-case regret of the ROBO problem. In the ROBO, the expectation of the regret on loss functions $\{f_1, f_2, \ldots, f_T\}$ is

$$\mathbb{E}_{f_1,\ldots,f_T}[R(T)] = \mathbb{E}_{f_1,\ldots,f_T}[\sum_{t=1}^{T} f_t(x_t) - \min_{x \in \mathcal{K}} \sum_{t=1}^{T} f_t(x)]$$

$$= \mathbb{E}_{f_1,\ldots,f_T}[\sum_{t=1}^{T} f_t(x_t)] - \mathbb{E}_{f_1,\ldots,f_T}[\min_{x \in \mathcal{K}} \sum_{t=1}^{T} f_t(x)]. \tag{9}$$

Since $f_t$ is uniformly and independently chosen in $\{f_+, f_-\}$, we get

$$\mathbb{E}_{f_1,\ldots,f_T}[\sum_{t=1}^{T} f_t(x_t)] = \sum_{t=1}^{T} \mathbb{E}_{f_t}[f_t(x_t)]$$

$$= \sum_{t=1}^{T} \frac{1}{2}(Lf_+(x_t) + Lf_-(x_t))$$

$$\ge \frac{LT}{2} \min_{x \in \mathcal{K}}(f_+(x) + f_-(x)).$$

From Lemma 5,

$$\mathbb{E}_{f_1,\ldots,f_T}[\sum_{t=1}^{T} f_t(x_t)] \geq 0. \tag{10}$$

Putting (10) into (9), we obtain

$$\mathbb{E}_{f_1,\ldots,f_T}[R(T)] \geq -\mathbb{E}_{f_1,\ldots,f_T}[\min_{x \in \mathcal{K}} \sum_{t=1}^{T} f_t(x)].$$

By Lemma 5,

$$\mathbb{E}_{f_1,\ldots,f_T}[R(T)] \geq -\mathbb{E}_{f_1,\ldots,f_T}[\min_{x \in \mathcal{K}} \sum_{t=1}^{T} f_t(x)]$$

$$= -\mathbb{E}_{f_1,\ldots,f_T}\left[ -DL| \sum_{f_t=Lf_+} 1 - \sum_{f_t=Lf_-} 1| \right]$$

$$= \mathbb{E}_{\epsilon_1,\ldots,\epsilon_T}\left[ DL| \sum_{\epsilon_t=1} 1 + \sum_{\epsilon_t=-1} -1| \right]$$

$$= \mathbb{E}_{\epsilon_1,\ldots,\epsilon_T}\left[ DL| \sum_{t=1}^{T} \epsilon_t| \right],$$

where $\epsilon_t$ are i.i.d Rademacher variables $\epsilon_t = \pm 1$ with probability 1/2. From the Khinchine's inequality [4], we finally get

$$\mathbb{E}_{f_1,\ldots,f_T}[R(T)] \geq \frac{DL}{\sqrt{2}}\mathbb{E}_{\epsilon_1,\ldots,\epsilon_T}\left[ \sum_{t=1}^{T} \epsilon_t^2 \right]^{\frac{1}{2}} = \frac{DL}{\sqrt{2}}\sqrt{T}, \tag{11}$$

which indicates that no matter how we choose strategies in the ROBO, there are a sequence of functions $\{f_1,\ldots,f_T\} \in \{Lf_+, Lf_-\}^T$ to make the regret no less than $\frac{DL}{\sqrt{2}}\sqrt{T}$. Considering that the diameter of the set $\mathcal{K}$ is $2D$ and the Lipschitz constant of $\{Lf_+, Lf_-\}$ is $L$, we complete our proof.

## C Proofs of Lemmas 1 and 2

In this appendix, we first introduce some fundamental definitions and technical lemmas, and then we prove Lemmas 1 and 2 in Subsections C.2 and C.3, respectively.

### C.1 Basic Definitions and Technical Lemmas

We discuss two special kinds of vector fields, namely, the *Killing field* and the *Jacobi field*.

**Definition 2** (Killing field). A vector field $\eta$ is a Killing field if it satisfies

$$\langle \nabla_X \eta, Y \rangle + \langle \nabla_Y \eta, X \rangle = 0, \forall X, Y \in \mathfrak{X}(M),$$

**Definition 3** (Jacobi field). A vector field $\eta$ along a geodesic $\gamma$ is a Jacobi field if it satisfies the Jacobi equation

$$\nabla_{\dot{\gamma}} \nabla_{\dot{\gamma}} \eta + R(\dot{\gamma}, \eta)\dot{\gamma} = 0,$$

where $R$ is the curvature tensor of $\mathcal{M}$ (see [5, Chapter 4.2]).

For vector fields, we mainly consider about their *flows* and *divergence*.

**Definition 4** (Flow). Suppose that $\mathcal{M}$ is a smooth manifold, $X \in \mathfrak{X}(\mathcal{M})$ and there is a smooth map $\phi : \mathbb{R} \times \mathcal{M} \to \mathcal{M}$,

$$\phi_t(p) = \phi(t, p), (t, p) \in \mathbb{R} \times \mathcal{M},$$

satisfying the following conditions:

1) $\phi_0(p) = p$;

2) $\phi_s \circ \phi_t = \phi_{s+t}$ for any real numbers $s, t$;

3) $X(p) = \frac{\partial \phi_t(p)}{\partial t}|_{t=0}$.

Then we call $\phi_t$ the flow (or the one-parameter group of diffeomorphism) of $X$, and call $X$ the infinitesimal transformation of $\phi_t$.

**Definition 5** (Divergence). For a vector field $X$, the divergence $Div(X)$ is the trace of the operator $\nabla X$. More precisely, if $\{e_1, \ldots, e_n\}$ is a normal orthogonal basis at tangent space $T_x \mathcal{M}$, the divergence of $X$ at $x$ can be expressed as

$$Div(X)(x) = \sum_{i=1}^{n} \langle \nabla_{e_i} X(x), e_i \rangle.$$

Then we present some lemmas that are needed in the following proofs.

**Lemma 6** ([2]). *If $\eta \in \mathfrak{X}(M)$ is a Killing vector field, then*

1) *For every vector field $X$, $\langle \nabla_X \eta, X \rangle = 0$. As a corollary, the divergence $Div(\eta) \equiv 0$.*

2) *For every geodesic $\gamma$, $\eta|_\gamma$ is a Jacobi field.*

**Lemma 7** ([3]). *If $\eta$ is a Jacobi vector field along a geodesic $\gamma : [0,1] \to \mathbb{R}$. Denote $\eta(t) = \eta(\gamma(t))$. Then*
$$\langle \eta(t), \dot{\gamma}(t) \rangle = \langle \eta(0), \dot{\gamma}(t) \rangle + t \langle \nabla_{\dot{\gamma}} \eta(0), \dot{\gamma}(t) \rangle$$
*for all $t \in [0,1]$.*

**Lemma 8** ([7]). *Let $\mathcal{M}$ be a simply connected complete Riemannian homogeneous manifold. Then for every $x \in M$ and every $X \in T_x \mathcal{M}$, there exists a Killing vector field $\eta$ such that $\eta(x) = X$. The flow of $\eta$ exists and consists of a one-parameter group of isometries.*

**Lemma 9** (Divergence theorem,[6]). *Let $\mathcal{M}$ be a Riemannian manifold $\mathcal{M}$ with the volume form $\omega$, $\mathcal{K} \subset \mathcal{M}$ with the boundary $\partial \mathcal{K}$, and $\vec{n}$ be the (outer) unit normal vector field of $\partial K$. Then, for any vector field $X$ and any differentiable function $f$,*

$$\int_{\mathcal{K}} X(f)(u)\omega = \int_{\partial \mathcal{K}} f(u)\langle X, \vec{n} \rangle \omega_{\partial \mathcal{K}} - \int_{\mathcal{K}} Div(X) f(u)\omega,$$

*where $\omega_{\partial \mathcal{K}}$ is the volume form of $\partial \mathcal{K}$ induced by $\omega$.*

**Lemma 10** (Bishop-Gromov volume comparison theorem, [6]). *Let $\mathcal{M}$ be an $n$-dimensional Riemannian manifold with sectional curvature lower bounded by $-\kappa$ ($\kappa \geq 0$). Given $p \in \mathcal{M}$, denote $V_r$ as the volume of the ball of radius $\delta$ about $p$ and $V_{r,\kappa}$ as the volume of a ball of radius $r$ on the $n$-dimensional hyperbolic space with constant curvature $-\kappa$. Then the function*

$$g(r) = \frac{V_r}{V_{r,\kappa}}$$

*is non-increasing.*

## C.2 Proof of Lemma 1

We start with the first part of the lemma.

Take a vector $X \in M_x$ arbitrarily. From Lemma 8, we can find a Killing vector field $\eta$ on $\mathcal{M}$ such that $\eta(x) = X$. The flow of $\eta$ consists of a one-parameter group of isometries $\{\phi_t\}_{t \in \mathbb{R}}$. Then the directional derivative of $\hat{f}$ along $X$ can be written as

$$X(\hat{f}(x)) = \lim_{t \to 0} \frac{\hat{f}(\phi_t(x)) - \hat{f}(x)}{t} = \frac{1}{V_\delta} \lim_{t \to 0} \frac{1}{t} \left( \int_{B_\delta(\phi_t(x))} f(u)\omega - \int_{B_\delta(x)} f(u)\omega \right). \quad (12)$$

Since $\phi_t$ is an isometry that preserves the distance, $\phi_t(B_\delta(x)) = B_\delta(\phi_t(x))$. By the substitution rule of integration ([5, Chapter 3.3]), we have

$$\int_{B_\delta(\phi_t(x))} f(u)\omega = \int_{B_\delta(x)} f(\phi_t(u))\phi_t^*(\omega). \quad (13)$$

Because $\phi_t$ preserves the metric $g$, it preserves the volume form, i.e., $\phi_t^*(\omega) = \omega$, which gives

$$\int_{B_\delta(\phi_t(x))} f(u)\omega = \int_{B_\delta(x)} f(\phi_t(u))\omega. \tag{14}$$

Combining equations (12) and (14) together, we have

$$X(\hat{f}(x)) = \frac{1}{V_\delta} \int_{B_\delta(x)} \lim_{t\to 0} \frac{f(\phi_t(u)) - f(u)}{t}\omega$$

$$= \frac{1}{V_\delta} \int_{B_\delta(x)} (\frac{\partial \phi_t(p)}{\partial t}|_{t=0}) f\omega. \tag{15}$$

By Definition 4, $\frac{\partial \phi_t(u)}{\partial t}|_{t=0} = \eta(u)$. Hence, we rewrite (15) as

$$X(\hat{f}(x)) = \frac{1}{V_\delta} \int_{B_\delta(x)} \eta(f)\omega.$$

According to Lemma 9,

$$X(\hat{f}(x)) = \frac{1}{V_\delta} \int_{S_\delta(x)} f(u)\langle \eta(u), \vec{n}(u)\rangle \omega_{S_\delta(x)} - \frac{1}{V_\delta} \int_{B_\delta(x)} Div(\eta)(u) f(u)\omega$$

$$= \frac{1}{V_\delta} \int_{S_\delta(x)} f(u)\langle \eta(u), \vec{n}(u)\rangle \omega_{S_\delta(x)}, \tag{16}$$

where $\omega_{S_\delta(x)}$ is the volume form of $S_\delta(x)$ induced by $\omega$ and $\vec{n}$ is the (outer) unit normal vector field of $S_\delta(x)$. The last equation is because $Div(\eta) \equiv 0$, as stated in Lemma 6.

Then we compute $\langle \eta, \vec{n}\rangle$ for each point $u \in S_\delta(x)$. Since geodesics start at the center $x$ are normal to the sphere $S_\delta(x)$, the outer normal vector $\vec{n}(u)$ can be written as $\frac{\dot{\gamma}_u(1)}{\|\dot{\gamma}_u(1)\|}$ for the geodesic $\gamma_u$ such that $\gamma_u(0) = x$ and $\gamma_u(1) = u$. Therefore,

$$\langle \eta(u), \vec{n}(u)\rangle = \frac{1}{\|\dot{\gamma}_u(1)\|} \langle \eta(\gamma_u(1)), \dot{\gamma}_u(1)\rangle.$$

Since $\eta$ is Killing, by Lemma 6, $\eta(\gamma_u(t))$ is Jacobi. By Lemma 7,

$$\langle \eta(u), \vec{n}(u)\rangle = \frac{1}{\|\dot{\gamma}_u(1)\|} \langle \eta(\gamma_u(1)), \dot{\gamma}_u(1)\rangle$$

$$= \frac{1}{\|\dot{\gamma}_u(1)\|} \langle \eta(\gamma_u(0)), \dot{\gamma}_u(0)\rangle + 1\langle \nabla_{\dot{\gamma}_u} \eta(\gamma_u(0)), \dot{\gamma}_u(0)\rangle$$

$$= \frac{1}{\|\dot{\gamma}_u(0)\|} \langle \eta(\gamma_u(0)), \dot{\gamma}_u(0)\rangle + 0. \tag{17}$$

Applying $\eta(\gamma_u(0)) = \eta(x) = X$ and $\dot{\gamma}_u(0) = \exp_x^{-1}(u)$ to (17) yields

$$\langle \eta(u), \vec{n}(u)\rangle = \frac{\langle X, \exp_x^{-1}(u)\rangle}{\|\exp_x^{-1}(u)\|}. \tag{18}$$

Substituting (18) to (16), we have

$$X(\hat{f}(x)) = \frac{1}{V_\delta} \int_{S_\delta(x)} f(u)\frac{\langle X, \exp_x^{-1}(u)\rangle}{\|\exp_x^{-1}(u)\|}\omega_{S_\delta(x)} = \langle \frac{1}{V_\delta} \int_{S_\delta(x)} f(u)\frac{\exp_x^{-1}(u)}{\|\exp_x^{-1}(u)\|}\omega_{S_\delta(x)}, X\rangle.$$

Because the directional derivative $X(\hat{f}(x))$ coincides with the term $\langle \nabla\hat{f}(x), X\rangle$, we obtain

$$\langle \nabla\hat{f}(x), X\rangle = \langle \frac{1}{V_\delta} \int_{S_\delta(x)} f(u)\frac{\exp_x^{-1}(u)}{\|\exp_x^{-1}(u)\|}\omega_{S_\delta(x)}, X\rangle.$$

Since $X$ is arbitrary,

$$\nabla\hat{f}(x) = \frac{1}{V_\delta} \int_{S_\delta(x)} f(u)\frac{\exp_x^{-1}(u)}{\|\exp_x^{-1}(u)\|}\omega_{S_\delta(x)} = \frac{S_\delta}{V_\delta} E_{u\in S_\delta(x)}\left[f(u)\frac{\exp_x^{-1}(u)}{\|\exp_x^{-1}(u)\|}\right],$$

which completes the proof of the first part.

Then we examine the second part of the lemma.

From the first part, it is clearly to see $\|\nabla \hat{f}(x)\| \leq \frac{S_\delta}{V_\delta} C$. Since the sectional curvature of $\mathcal{M}$ is lower bounded by $-\kappa$, the function $g(r) = \frac{V_r}{V_{r,\kappa}}$ is non-increasing from Lemma 10 and so does $\log g(r)$. Therefore,

$$\frac{d}{dr} log(g(r)) = \frac{d}{dr} \log V_r - \frac{d}{dr} \log V_{r,\kappa} \leq 0.$$

Since deriving the volume of a ball along the radius gives the surface area of its sphere, we write

$$\frac{d}{dr} log(g(r)) = \frac{S_r}{V_r} - \frac{S_{r,\kappa}}{V_{r,\kappa}} \leq 0, \tag{19}$$

where $S_r$ and $S_{r,\kappa}$ are the surface area of the balls in $\mathcal{M}$ and the hyperbolic space, respectively.

Setting $r = \delta$ in (19), we get $\frac{S_\delta}{V_\delta} \leq \frac{S_{\delta,\kappa}}{V_{\delta,\kappa}}$. From calculation, it shows that

$$\frac{S_{\delta,\kappa}}{V_{\delta,\kappa}} = \frac{\sinh^{n-1}(\sqrt{\kappa}\delta)}{\int_0^\delta \sinh^{n-1}(\sqrt{\kappa}t)dt}.$$

Consequently,

$$\|\nabla \hat{f}(x)\| \leq C \frac{\sinh^{n-1}(\sqrt{\kappa}\delta)}{\int_0^\delta \sinh^{n-1}(\sqrt{\kappa}t)dt}.$$

By a change of variable $u = \sinh t$,

$$\int_0^\delta \sinh^{n-1}(\sqrt{\kappa}t)dt = \kappa^{-1/2} \int_0^{\sinh(\sqrt{\kappa}\delta)} u^{n-1}(1+u)^{-1/2}du.$$

Integration by parts gives

$$\int_0^\delta \sinh^{n-1}(\sqrt{\kappa}t)dt = \frac{\sinh^n(\sqrt{\kappa}\delta)}{n\sqrt{\kappa}\cosh(\sqrt{\kappa}\delta)} + \kappa^{-1/2} \int_0^{\sinh(\sqrt{\kappa}\delta)} u^{n-1}(1+u)^{-1/2}du$$

$$\geq \frac{\sinh^n(\sqrt{\kappa}\delta)}{n\sqrt{\kappa}\cosh(\sqrt{\kappa}\delta)}.$$

Putting it into the expression of $\frac{S_\delta}{V_\delta}$, we get

$$\frac{S_\delta}{V_\delta} \leq n\sqrt{\kappa}\coth(\sqrt{\kappa}\delta).$$

Applying the inequality $\coth(x) < x + 1/x$, we have

$$\frac{S_\delta}{V_\delta} \leq \frac{n}{\delta} + n\kappa\delta, \quad \forall \delta > 0.$$

Hence, for every $\delta > 0$,

$$\|\nabla \hat{f}(x)\| \leq \frac{S_\delta}{V_\delta} C \leq C\left(\frac{n}{\delta} + n\kappa\delta\right),$$

which completes our proof.

### C.3  Proof of Lemma 2

Without loss of generality, we assume $f(x) = 0$. By the homogeneity of the manifold $\mathcal{M}$, we find an isometry $\phi$ such that $\phi(x) = y$. Denote $V(u)$ as the vector field $V(u) = \exp_u^{-1}(\phi(u))$. Clearly,

$$\hat{f}(y) - \hat{f}(x) = \frac{1}{V_\delta}\left(\int_{B_\delta(y)} f(u)\omega - \int_{B_\delta(x)} f(u)\omega\right)$$

$$= \frac{1}{V_\delta}\left(\int_{B_\delta(\phi(x))} f(u)\omega - \int_{B_\delta(x)} f(u)\omega\right).$$

With the method shown in (13) and (14),

$$\hat{f}(y) - \hat{f}(x) = \frac{1}{V_\delta} \int_{B_\delta(x)} f(\phi(u)) - f(u)\omega.$$

By the g-convexity of $f$,

$$\hat{f}(y) - \hat{f}(x) = \frac{1}{V_\delta} \int_{B_\delta(x)} f(\phi(u)) - f(u)\omega$$

$$\geq \frac{1}{V_\delta} \int_{B_\delta(x)} \langle \nabla f(u), \exp_u^{-1}(\phi(u)) \rangle \omega$$

$$= \frac{1}{V_\delta} \int_{B_\delta(x)} \langle \nabla f(u), V(u) \rangle \omega$$

$$= \frac{1}{V_\delta} \int_{B_\delta(x)} V(f(u))\omega. \tag{20}$$

By Lemma 9,

$$\int_{B_\delta(x)} V(f(u))\omega = \int_{S_\delta(x)} f(u)\langle V(u), \vec{n}(u) \rangle \omega_{S_\delta(x)} - \int_{B_\delta(x)} Div(V) f(u)\omega. \tag{21}$$

Hence, we rewrite (20) as

$$\hat{f}(y) - \hat{f}(x) \geq \frac{1}{V_\delta} \left( \int_{S_\delta(x)} f(u)\langle V(u), \vec{n}(u) \rangle \omega_{S_\delta(x)} - \int_{B_\delta(x)} Div(V) f(u)\omega \right). \tag{22}$$

In Lemma 1, we have already shown that

$$\langle \nabla \hat{f}(x), \exp_x^{-1}(y) \rangle = \langle \frac{1}{V_\delta} \int_{S_\delta(x)} f(u) \frac{\exp_x^{-1}(u)}{\|\exp_x^{-1}(u)\|} \omega_{S_\delta(x)}, \exp_x^{-1}(y) \rangle$$

$$= \langle \frac{1}{V_\delta} \int_{S_\delta(x)} f(u) \frac{\exp_x^{-1}(u)}{\|\exp_x^{-1}(u)\|} \omega_{S_\delta(x)}, V(x) \rangle \tag{23}$$

Denote $\vec{m}(u)$ as the vector $\frac{\exp_x^{-1}(u)}{\|\exp_x^{-1}(u)\|}$. Combining (21) and (23) gives

$$\hat{f}(y) - \hat{f}(x) - \langle \nabla \hat{f}(x), \exp_x^{-1}(y) \rangle \geq \frac{1}{V_\delta} \left( \int_{S_\delta(x)} f(u) \Big( \langle V(u), \vec{n}(u) \rangle - \langle V(x), \vec{m}(u) \rangle \Big) \omega_{S_\delta(x)} \right)$$

$$- \frac{1}{V_\delta} \left( \int_{B_\delta(x)} Div(V) f(u)\omega \right). \tag{24}$$

Here we claim

$$\langle V(u), \vec{n}(u) \rangle - \langle V(x), \vec{m}(u) \rangle \leq 0, \quad \forall u \in S_\delta(x). \tag{$*$}$$

If the claim $(*)$ holds (whose proof will be presented in C.4), then, with the g-$L$-Lipschitz of $f$ and the condition $f(x) = 0$, we have

$$\int_{S_\delta(x)} f(u) \Big( \langle V(u), \vec{n}(u) \rangle - \langle V(x), \vec{m}(u) \rangle \Big) \omega_{S_\delta(x)}$$

$$\geq \int_{S_\delta(x)} \delta L \Big( \langle V(u), \vec{n}(u) \rangle - \langle V(x), \vec{m}(u) \rangle \Big) \omega_{S_\delta(x)}$$

$$= \left( \int_{S_\delta(x)} \delta L \langle V(u), \vec{n}(u) \rangle \omega_{S_\delta(x)} \right) - \left( \int_{S_\delta(x)} \delta L \langle V(x), \vec{m}(u) \rangle \omega_{S_\delta(x)} \right). \tag{25}$$

By Lemma 1, $\frac{1}{V_\delta} \int_{S_\delta(x)} \delta L \langle V(x), \vec{m}(u) \rangle \omega_{S_\delta(x)}$ in (25) is the gradient of the function

$$\hat{g}(x) = \frac{1}{V_\delta} \left( \int_{B_\delta(x)} \delta L \cdot \omega \right) \equiv \delta L,$$

and then

$$\frac{1}{V_\delta} \int_{S_\delta(x)} \delta L \langle V(x), \vec{m}(u) \rangle \omega_{S_\delta(x)} = 0. \tag{26}$$

Combining (24)-(26), we have

$$\hat{f}(y) - \hat{f}(x) - \langle \nabla \hat{f}(x), \exp_x^{-1}(y) \rangle \geq \frac{1}{V_\delta} \left( \int_{S_\delta(x)} \delta L \langle V(u), \vec{n}(u) \rangle \omega_{S_\delta(x)} \right)$$
$$- \frac{1}{V_\delta} \left( \int_{B_\delta(x)} Div(V) f(u) \omega \right).$$

Applying Lemma 9 again, we obtain

$$\frac{1}{V_\delta} \int_{S_\delta(x)} \delta L \langle V(u), \vec{n}(u) \rangle \omega_{S_\delta(x)} = \frac{1}{V_\delta} \int_{B_\delta(x)} V(\delta L) \omega - \frac{1}{V_\delta} \left( \int_{B_\delta(x)} Div(V) \delta L \omega \right)$$
$$= -\frac{1}{V_\delta} \left( \int_{B_\delta(x)} Div(V) \delta L \omega \right).$$

Therefore,

$$\hat{f}(y) - \hat{f}(x) - \langle \nabla \hat{f}(x), \exp_x^{-1}(y) \rangle \geq -\frac{1}{V_\delta} \left( \int_{B_\delta(x)} Div(V)(f(x) + \delta L) \omega \right)$$
$$\geq -2\delta L \sup_{u \in B_\delta(x)} |Div(V(u))|. \tag{27}$$

Note that $V(u) = \exp_u^{-1}(\phi(u))$ is continuous on $p$ and $\phi$, and $\phi$ is continuous on $x$ and $y$. Thus, $|Div(V(u))|$ is a continuous function of $(x, y, u) \in \bar{\mathcal{K}} \times \bar{\mathcal{K}} \times \bar{\mathcal{K}}$. Denote

$$\rho = \sup_{(x,y,u) \in \bar{\mathcal{K}} \times \bar{\mathcal{K}} \times \bar{\mathcal{K}}} |Div(V(u))|.$$

Since the boundedness of $\mathcal{K}$ set yields the compactness of $\bar{\mathcal{K}} \times \bar{\mathcal{K}} \times \bar{\mathcal{K}}$, we have $\rho < \infty$. Putting $\rho$ into (27) establishes the formula in Lemma 7.

### C.4  Proof of the Claim (∗)

Fix $u \in S_\delta(x)$ and denote $\xi_u(s) = \exp_x(s\vec{m}(u))$ as the geodesic with the initial tangent vector $\vec{m}(u)$. Consider the following rectangle map

$$\Gamma_u : [0, 1] \times [0, \delta] \to \mathcal{M}$$
$$(t, s) \to \exp_{\xi_u(s)}(tV(\xi_u(s))).$$

Set $T(t, s) = \frac{\partial \Gamma_u}{\partial t}(t, s)$ and $S(t, s) = \frac{\partial \Gamma_u}{\partial s}(t, s)$. For a fixed $t$, the length of the curve $\gamma_t(s) = \Gamma_u(t, s), (0 \leq s \leq \delta)$ is defined as

$$l_u(t) = \int_0^\delta \sqrt{\langle S(t, s), S(t, s) \rangle} ds.$$

The first variation formula (see [6, Theorem 6.3]) gives,

$$l_u'(0) = \langle T(0, \delta), S(0, \delta) \rangle - \langle T(0, 0), S(0, 0) \rangle.$$

Because

$$T(0, s) = V(\xi_u(s)), \forall s \in [0, \delta]$$

and

$$S(0, 0) = \vec{m}(u), S(0, \delta) = \vec{n}(u),$$

we have

$$l_u'(0) = \langle V(u), \vec{n}(u) \rangle - \langle V(x), \vec{m}(u) \rangle.$$

To prove $(*)$, it is sufficient to show that $l_u'(0) \leq 0$. Let us focus on the second derivative of the function $l_u(t)$, that is,

$$l_u''(t) = \frac{d^2}{dt^2} \int_0^\delta \sqrt{\langle S(t,s), S(t,s) \rangle} ds$$

$$= \int_0^\delta \frac{d^2}{dt^2} \sqrt{\langle S(t,s), S(t,s) \rangle} ds$$

$$= \int_0^\delta \frac{d}{dt} \left( \frac{1}{\|S\|} \langle \nabla_T S, S \rangle \right) ds$$

$$= \int_0^\delta -\frac{1}{\|S\|^3} \langle \nabla_T S, S \rangle^2 + \frac{1}{\|S\|} \langle \nabla_T S, \nabla_T S \rangle + \frac{1}{\|S\|} \langle \nabla_T \nabla_T S, S \rangle ds. \quad (28)$$

For every fixed $s$, the curve $\gamma_s(t) = \Gamma_u(t,s)$ is a geodesic, hence, $S$ is the variation field of the geodesic $\gamma_s(t)$ and becomes a Jacobi field. Putting the Jaboci equation (Definition 4) into (28), we have

$$l_u''(t) = \int_0^\delta -\frac{1}{\|S\|^3} \langle \nabla_T S, S \rangle^2 + \frac{1}{\|S\|} \langle \nabla_T S, \nabla_T S \rangle + \frac{1}{\|S\|} - R(T, S, S, T) ds.$$

By the Cauchy–Schwarz inequality, $-\langle \nabla_T S, S \rangle^2 \geq -\|S\|^2 \|\nabla_T S\|^2$, which yields

$$l_u''(t) \geq \int_0^\delta -\frac{1}{\|S\|^3} - \|S\|^2 \|\nabla_T S\|^2 + \frac{1}{\|S\|} \langle \nabla_T S, \nabla_T S \rangle + \frac{1}{\|S\|} - R(T, S, S, T) ds$$

$$\geq \int_0^\delta \frac{1}{\|S\|} - R(T, S, S, T) ds$$

From the definition of the sectional curvature, $R(T, S, S, T) = K(\Pi)|T \wedge S|^2$, where $K(\Pi)$ is the sectional curvature of the two-dimensional submanifold spanned by $T$ and $S$. Since $\mathcal{M}$ has nonpostive sectional curvature, we get

$$l_u''(t) \geq \int_0^\delta \frac{1}{\|S\|} - R(T, S, S, T) ds \geq 0,$$

which means that $l_u(t)$ is convex in $[0,1]$.

Let us look back on the function $l_u(t)$. Note that the 0-curve is

$$\gamma_s(0) = \xi(s),$$

and the 1-curve is

$$\gamma_s(1) = \exp_{\xi(s)}(V(\xi(s))) = \exp_{\xi(s)}(\exp_{\xi(s)}^{-1}(\phi(\xi(s))) = \phi(\xi(s)).$$

Since the mapping $\phi$ is an isometry, the length of $\xi(s)$ is equal to the length of $\phi(\xi(s))$. As a result,

$$l_u(0) = l_u(1).$$

The convexity of $l_u$ immediately leads to

$$l_u'(0) \leq 0,$$

which proves the claim $(*)$.

## D   Proof of Theorem 4

Before the proof, we propose two lemmas. Lemma 11 is about the expected online gradient descent on Riemannian manifolds.

**Lemma 11.** *Suppose that $S$ is a g-convex set of $\mathcal{M}$ with diameter $D$ and $\{f_t\}_{t=1,2,\dots,T}$ is a series of smooth functions and there exists a constant $\lambda \geq 0$ such that*

$$f_t(x) - f_t(y) - \langle \nabla f_t(x), \exp_x^{-1}(y) \rangle \geq -\lambda, \quad (29)$$

*for any $x, y \in S$ and $t = 1, 2, \ldots, T$. If the sequence $\{x_t\}_{t=1,2,\ldots,T}$ is generated by*

$$x_{t+1} = P_S(\exp_{x_t}(-\alpha g_t)),$$

*where $\alpha > 0$ and $g_t$ is a random vector bounded by $G$ such that $\mathbb{E}[g_t | x_t] = \nabla f_t(x_t)$ for every $t = 1, 2, \ldots, T$, then, with taking $\alpha = \frac{D}{G\sqrt{\zeta(\kappa, D)T}}$, we have*

$$\mathbb{E}\Big[\sum_{t=1}^{T} f_t(x_t)\Big] - \min_{x \in \mathcal{K}} \sum_{t=1}^{T} f_t(x) \leq DG\sqrt{\zeta(\kappa, D)T} + \lambda T.$$

*Proof.* Let $x^* = \arg\min_{x \in \mathcal{K}} \sum_{t=1}^{T} f_t(x)$. From (29), the difference between $f_t(x_t)$ and $f_t(x^*)$ is bounded by

$$
\begin{aligned}
f_t(x_t) - f_t(x^*) &\leq \langle \nabla f_t(x_t), \exp_{x_t}^{-1}(x^*)\rangle + \lambda \\
&= \langle \mathbb{E}[g_t | x_t], \exp_{x_t}^{-1}(x^*)\rangle + \lambda \\
&= \mathbb{E}\Big[\langle g_t, \exp_{x_t}^{-1}(x^*)\rangle | x_t\Big] + \lambda.
\end{aligned}
$$

Taking the expectation on both sides yields

$$\mathbb{E}[f_t(x_t) - f_t(x^*)] \leq \mathbb{E}\Big[\langle g_t, \exp_{x_t}^{-1}(x^*)\rangle\Big] + \lambda.$$

From Lemma 3,

$$\mathbb{E}[f_t(x_t) - f_t(x^*)] \leq \mathbb{E}\Big[\frac{1}{2\alpha}(d^2(x_t, x^*) - d^2(x_{t+1}, x^*)) + \frac{1}{2}\zeta(\kappa, d(x_t, x^*))\alpha\|g_t\|^2\Big] + \lambda. \quad (30)$$

Summing (30) from 1 to $T$, we have

$$
\begin{aligned}
\sum_{t=1}^{T} \mathbb{E}[f_t(x_t) - f_t(x^*)] &\leq \sum_{t=1}^{T} \mathbb{E}\Big[\frac{1}{2\alpha}(d^2(x_t, x^*) - d^2(x_{t+1}, x^*)) + \frac{\alpha}{2}\zeta(\kappa, d(x_t, x^*))\|g_t\|^2\Big] + \lambda T \\
&\leq \mathbb{E}\Big[\frac{1}{2\alpha}d^2(x_1, x^*)\Big] + \sum_{t=1}^{T} \mathbb{E}\Big[\frac{\alpha}{2}\zeta(\kappa, d^2(x_t, x^*))G^2\Big] + \lambda T \\
&\leq \frac{D^2}{2\alpha} + \frac{\alpha}{2}\zeta(\kappa, D)G^2 T + \lambda T. \quad (31)
\end{aligned}
$$

The last inequality is because $S$ is of diameter $D$. Putting $\alpha = \frac{D}{G\sqrt{\zeta(\kappa, D)T}}$ in (31), we complete our proof. $\qquad\square$

Lemma 12 reveals a relationship between the offline optimum in $(1 - \tau)\mathcal{K}$ and $\mathcal{K}$.

**Lemma 12.** *Suppose that Assumption 6 holds, and $\{f_t\}_{t=1,\ldots,T}$ is a sequence of g-convex function defined on $\mathcal{K}$ bounded by $C$. Then*

$$\min_{x \in (1-\tau)\mathcal{K}} \sum_{t=1}^{T} f_t(x) \leq 2\tau CT + \min_{x \in \mathcal{K}} \sum_{t=1}^{T} f_t(x)$$

*Proof.* Since $(1 - \tau)\mathcal{K} = \{\exp_p((1 - \tau)u) | u = \exp_p^{-1}(x) \in \mathcal{K}\}$, it is easy to check

$$\min_{x \in (1-\tau)\mathcal{K}} \sum_{t=1}^{T} f_t(x) = \min_{x \in \mathcal{K}} \sum_{t=1}^{T} f_t\Big(\exp_p((1 - \tau)\exp_p^{-1}(x))\Big).$$

By the g-convexity of $f_t$, we have

$$\min_{x \in \mathcal{K}} \sum_{t=1}^{T} f_t\Big(\exp_p((1-\tau)\exp_p^{-1}(x))\Big) \leq \min_{x \in \mathcal{K}} \sum_{t=1}^{T} \tau f_t(p) + (1-\tau) f_t(x)$$

$$\leq \min_{x \in \mathcal{K}} \sum_{t=1}^{T} \tau(f_t(p) - f_t(x)) + f_t(x)$$

$$\leq \min_{x \in \mathcal{K}} \sum_{t=1}^{T} \tau 2C + f_t(x)$$

$$\leq 2\tau CT + \min_{x \in \mathcal{K}} \sum_{t=1}^{T} f_t(x),$$

which completes our proof. $\qquad\square$

Now it is time to prove Theorem 4.

*Proof of Theorem 4.* Denote $x_\tau^*$ as the minimizer of the problem $\min_{x \in (1-\tau)\mathcal{K}} \sum_{t=1}^{T} f_t(x)$, and then the expectation can be reformulated as

$$\mathbb{E}[R(T)] = \sum_{t=1}^{T} \mathbb{E}\Big[ f_t(x_t) - f_t(x^*) \Big]$$

$$= \mathbb{E}\Big[ \sum_{t=1}^{T} (f_t(x_t) - f_t(y_t)) \Big] + \mathbb{E}\Big[ \sum_{t=1}^{T} (f_t(y_t) - \hat{f}_t(y_t)) \Big] + \mathbb{E}\Big[ \sum_{t=1}^{T} (\hat{f}_t(y_t) - \hat{f}_t(x_\tau^*)) \Big]$$

$$+ \mathbb{E}\Big[ \sum_{t=1}^{T} (\hat{f}_t(x_\tau^*) - f_t(x_\tau^*)) \Big] + \mathbb{E}\Big[ \sum_{t=1}^{T} (f_t(x_\tau^*) - f_t(x^*)) \Big].$$

The Lipschitz condition leads to $|f_t(x_t) - f_t(y_t)| \leq \delta L$ and $|f_t(x) - \hat{f}_t(x)| < \delta L$, and recalling Lemma 12 yields $\sum_{t=1}^{T} (f_t(x_\tau^*) - f_t(x^*)) \leq 2\tau CT$. Putting them altogether,

$$\mathbb{E}[R(T)] \leq \mathbb{E}\Big[ \sum_{t=1}^{T} (\hat{f}_t(y_t) - \hat{f}_t(x_\tau^*)) \Big] + 3\delta LT + 2\tau CT \tag{32}$$

To estimate the term $\mathbb{E}\Big[ \sum_{t=1}^{T} (\hat{f}_t(y_t) - \hat{f}_t(x_\tau^*)) \Big]$, we focus on the update rule of $y_t$, that is,

$$y_{t+1} = P_{(1-\tau)\mathcal{K}}\left( \exp_{y_t}(\alpha f_t(x_t) \frac{\exp_{y_t}^{-1}(x_t)}{\|\exp_{y_t}^{-1}(x_t)\|}) \right)$$

$$= P_{(1-\tau)\mathcal{K}}\left( \exp_{y_t}(\frac{D}{C\sqrt{\zeta(\kappa,D)T}} f_t(x_t) \frac{\exp_{y_t}^{-1}(x_t)}{\|\exp_{y_t}^{-1}(x_t)\|}) \right)$$

$$= P_{(1-\tau)\mathcal{K}}\left( \exp_{y_t}(\frac{D}{\frac{S_\delta}{V_\delta} C\sqrt{\zeta(\kappa,D)T}} \frac{S_\delta}{V_\delta} f_t(x_t) \frac{\exp_{y_t}^{-1}(x_t)}{\|\exp_{y_t}^{-1}(x_t)\|}) \right).$$

From what we have proved in Lemma 1 we obtain $\mathbb{E}\Big[ \frac{S_\delta}{V_\delta} f_t(x_t) \frac{\exp_{y_t}^{-1}(x_t)}{\|\exp_{y_t}^{-1}(x_t)\|} \Big| y_t \Big] = \nabla \hat{f}(y_t)$ and $\|\hat{f}(y_t)\| \leq \frac{S_\delta}{V_\delta} C$. Consequently, it is clear to see that the update rule here is that of expected gradient descent in Lemma 11 with parameters $S = (1-\tau)\mathcal{K}$, $G = \frac{S_\delta}{V_\delta} C$, $\lambda = 2\delta \rho L$ and the step size $\alpha = \frac{D}{G\sqrt{\zeta(\kappa,D)T}}$. Thus,

$$\mathbb{E}\Big[ \sum_{t=1}^{T} f_t(x_t) \Big] - \min_{x \in (1-\tau)\mathcal{K}} \sum_{t=1}^{T} f_t(x) \leq \frac{S_\delta}{V_\delta} DC \sqrt{\zeta(\kappa,D)T} + 2\delta \rho LT. \tag{33}$$

Applying (33), we rewrite (32) as

$$\mathbb{E}[R(T)] \leq \frac{S_\delta}{V_\delta} DC\sqrt{\zeta(\kappa,D)T} + 3\delta LT + 2\tau CT + 2\delta\rho LT.$$

By Lemma 1,

$$\frac{S_\delta}{V_\delta} \leq \frac{n}{\delta} + n\kappa\delta = \frac{n}{\delta} + B\delta.$$

Then

$$\mathbb{E}[R(T)] \leq (\frac{n}{\delta} + B\delta)DC\sqrt{\zeta(\kappa,D)T} + 3\delta LT + 2\tau CT + 2\delta\rho LT. \tag{34}$$

Taking $\tau = \frac{\delta}{r}$, $\Delta = BCD\sqrt{\zeta(\kappa,D)} + 3L + 2C/r$ and $\delta = T^{-\frac{1}{4}}\sqrt{\frac{CDr\sqrt{\zeta(\kappa,D)}}{\Delta}}$, we have,

$$\begin{aligned}
\mathbb{E}[R(T)] &\leq \frac{n}{\delta}DC\sqrt{\zeta(\kappa,D)T} + B\delta DC\sqrt{\zeta(\kappa,D)T} + 3\delta LT + 2\tau CT + 2\delta\rho LT \\
&\leq \frac{n}{\delta}DC\sqrt{\zeta(\kappa,D)T} + (B\delta DC\sqrt{\zeta(\kappa,D)} + 3\delta L + 2\tau C)T + 2\delta\rho LT, \\
&= 2T^{\frac{3}{4}}\sqrt{nCD\sqrt{\zeta(\kappa,D)}}\left(\sqrt{\Delta} + \frac{2\rho L}{\sqrt{\Delta}}\right).
\end{aligned}$$

Also with $\Lambda = \sqrt{\Delta} + \frac{2\rho L}{\sqrt{\Delta}}$, we get

$$\mathbb{E}[R(T)] \leq 2T^{\frac{3}{4}}\sqrt{nCD\sqrt{\zeta(\kappa,D)}}\Lambda,$$

which completes our proof. $\square$