# OpenReview forum: "No-regret Online Learning over Riemannian Manifolds"
_NeurIPS.cc/2021/Conference — NeurIPS 2021 Poster_

### Official Review · Reviewer_6h8i · 2021-06-30

**Rating:** 6
**Confidence:** 4

**Summary:**

This paper takes online learning algorithms designed for the Euclidean space (algorithms for online convex optimization with convex or strongly convex functions in the full information case. Or the convex case with bandit feedback) and generalizes them so they work in Hadamard manifolds with the corresponding notions of geodesic convexity. In the bandit case the manifolds are also assumed to be homogeneous.

A lower bound of \Omega(\sqrt{T}) on the expected regret is provided.

The regret bounds are greater than those in the Euclidean counterparts and in the bandit case (the main contribution) they are greater by a factor that depends on the dimension and that is not given in the paper ( its value is not worked out). This fact makes the claim in the abstract "All the obtained regret bounds match the corresponding results in Euclidean spaces." be false and misleading, in my opinion.

**Limitations And Societal Impact:**

Yes

**Main Review:**

**Comments on Theoretical results**

The lower bound proceeds analogously to the Euclidean case. The contribution of this lower bound is on noting the Busemann functions can be used with the lower bound technique coming from the Euclidean case.

The proofs in the full information setting are a simple consequence of the existing theory and the geometric lemma of Zhang and Sra.  The regret in this setting is increased (with respect to the Euclidean case) by a factor of a natural geometric constant depending on the diameter of the optimization set and a lower bound on the sectional curvature of the manifold. The analysis is an immediate consequence of the Euclidean analysis and a previously given geometric lemma in Zhang and Sra 2016 (actually the analysis is essentially the same as the analysis of Theorem 9 and Theorem 11 in that paper).


The main contribution of this work is, in my view, the algorithm for the bandit case, which requires more geometric machinery to be obtained. The final pseudoregret upper bound worsens with respect to the Euclidean case by a factor that contains some "constants" that among other things, depend on the dimension. (there is also the detail that for their bound to be true T must be quite large, but I consider this to be a less important detail). This is a nice result, but I think that regardless of the final value, a bound on those "constants" should be provided.

**Some important "constants" are not worked out: $\\epsilon(n)$ and $\\rho$**

Their magnitude should be provided in the form of a bound. \\rho appears in the regret and \\epsilon(n) appears in the lower bound required for T by the regret theorem. The "constants" depend on the dimension, bound on the sectional curvature and diameter, and the paper without providing a magnitude of these constants is incomplete.


**Mathematical typos (not serious, I believe)**

In Table 1: the lower bounds should use \Omega

Pseudocode of Algorithm 1 is not using the learning rates provided as input.

In the equalities before (27), after 'Applying Lemma 9 again, we obtain', after the first equality I believe one should be adding the 1/V_\delta times the integral of the divergence (as opposed to substracting). I believe the same conclusion reached in (27) can be reached though.

In the last page of the appendix, I believe there is a \kappa missing in the second summand of the rhs of the first inequality (and then the error carries forward) after 'Moreover, we get'

Also, the last equality of that set of lines seems to have the wrong constants. In particular, the r disappears despite of it being part of \delta. And the 2 that is in front of T^{3/4} seems to belong to the \sqrt{\Delta} inside the parentheses.


**Experiments**

The learning rate in the hyperbolic case seems to have been taken constant. Which I guess it is fine since the aim is to showcase the applicability of the method, but it should be reported in the main paper.

About the learning rate in the SPD, I might be wrong, but I think L should be equal to D in this case and therefore it should be \\eta_t = \\sqrt{tanh(D\\kappa)/(D\\kappa t)}, which is not the same thing that appears in the code


**Other comments**

In the comparison with related work it is said that "[12], [37], [38], [36]  assume prior knowledge of loss functions which is different from the framework of our R-OCO". While this is true, and as pointed out above, the analysis of some theorems in 37 is almost identical to the analysis of the full information case in this paper, while you seem to be implying they are very different things. This is essentially because the analysis provided is a mirror descent like analysis (but working with the \\ell_2 norm only)

Other works have made other uses of mirror descent analyses in Riemannian manifolds, and though they apply to the offline setting, there are connections between those and the online learning setting. For instance the paper "Global Riemannian Acceleration in Hyperbolic and Spherical Spaces" defines a mirror descent type of method and combines it with a gradient descent method to obtain acceleration.

l277 while it is true that f_t is 2-strongly convex, a citation to such a result should be provided, since it is not obvious a priori this is the case for someone that does not work in the field.



**Conclusion**

While I think the paper is interesting and of value, I think it needs some changes before it can be published. The magnitude of the "constants" \epsilon(n) and \rho should be worked out. There are some mathematical typos that need to be fixed.


---

There are other non-math typos that I list below:

The Assumption environment contains a typo so all assumptions say "Assmuption"

l41 'can be much expensive' rephrase

l88 'However, These studies' typo in capitalization

l122 'A function...' -> 'A differentiable function'

l143 'with the diameter D' ->  'with diameter D'

l163 'Theorems 1 and 2 also reveals' -> 'Theorems 1 and 2 also reveal'

l179 'In the following of this section' -> In the rest of this section


Algorithm 2. The for has two 'do's

l203 In equation (2) it should be \tilde{g}_t instead of \tilde{g_t}

l206 'expected gradient descent method' Do you mean stochastic gradient descent method? If not, please clarify or provide a citation that defines expected gradient method

l220 space missing after '(2)'. Note that in l213 there is a space after '(1)'

l224 'qudratic' typo

l235 'The first part of the lemma proceeds the result in [18]' rephrase

l237 'that transform' ->  'that transforms'

l256 '5-7 holds' -> '5-7 hold'

l443 'lisence' -> 'license'


The lines in the appendix should be numbered.
In the equation below (5) there is f(x^\star) that should be f_t(x^\star). Same for the equation after (6)

'by the strongly convexity' -> 'by the strong convexity'

in the second line of the inequality preceding (11), there is an L that should be a 1

Lemma 10 'Let M be a n-dimensional' ->  'Let M be an n-dimensional'

'is consist of' -> 'consists of'

In (19) the first V_r should have (x) not (\kappa)

proof of lemma 2 'without the loss of generality' ->  'without loss of generality'

after (25). It says it is by Lemma 6 but it is by Lemma 1

'and become a Jacobi field' -> 'and becomes a Jacobi field'

After 'Let us look back on the function', there is \gamma_{0}(s). This notation is not consistent with the notation used before, that had the s in the subscript.

In the equation below that one you have \gamma_1(s) = exp_{\xi(s)}(V(s)) but it should be \gamma_1(s) = exp_{\xi(s)}(V(\xi(s)))

lemma 11 'with the diameter D' -> 'with diameter D' ; Also 'are a series of smooth functions and there exists' ->  'be a series of smooth functions and assume there exists'

page 25 'To estimate the term E' E is not using mathbb. Same at the end of the page


**Time Spent Reviewing:**

12

---

> ### Author Response · Authors · 2021-08-10
> **Authors' response**
>
> We would like to thank you for the careful reading and the insightful suggestions. The mathematical typos that you pointed out will be carefully addressed in the final version, which greatly helped us improve the quality of the presentation.
>
> 	 Q1: Main contribution of the paper.
>
> In fact the R-OGD algorithm is not a simple consequence of [37]. Although the fundamental law of cosines established in [37] (which has been seen in quite a few works in Riemannian optimization) is utilized in this paper, our work is quite different due to the following technical advances:
> 1. Our work  concentrates on the Riemannian online optimization, which is to optimize time-varying functions without prior knowledge, rather than one deterministic, known function.
> 2. Our proposed R-OGD algorithm presents a new scheme for handling online time-varying functions on Hadamard manifolds. In addition to optimization, the dynamics of the algorithm predicts the next round strategy based on the existing information.
> 3. To illustrate tightness of the R-OGD algorithm, we derive analysis on the regret lower bound , where a novel construction with Busemann functions can overcome the obstacles of the absence of linear structure on manifolds.
>
> Therefore, we believe our results on online gradient descent over Riemannian manifolds still contain several non-trivial developments. In the meantime, we will highlight the fact that our analysis for full informaiton feedback case is inspired by and follows the same idea of the work of [37] and related works, in the final version of the paper.
>
> Moreover, as you pointed out, we acknowledge that the R-BAN algorithm is far more challenging, and may be the main contribution of our work.  The gradient estimator is hard to compute due to the lack of commutativity between the gradient operator and the integration operator; the bound of gradient estimator is difficult to estimate because of the geometry of $B_\delta(x)$; and the convexity of smoothed loss function may be violated.
>
> 	Q2: Constant magnitudes in regret bounds.
>
> We appreciate your comments on the importance of the constant magnitudes in the derived regret bounds, which indeed concerns whether the presented results are complete or not.  Here we would like to the constant magnitudes in our regret bounds are in fact clear.
>
>
>  * The constant $\rho$ can be expressed as
> $$\rho = \sup_{x,y,u\in K} |\frac{1}{\sqrt{G}} \frac{\partial{}}{\partial{x_i}}\big(\sqrt{G} \exp^{-1}_u \phi(u)\big)^i| \quad s.t. \quad \phi(x) = y  $$
> Hence, for a certain manifold $M$, once the set $K$ is fixed and the explicit expression of $\phi$ is given, we can compute the constant $\rho$ as a finite number. For example, for a $2$-dimensional Poincar\'e disk, the isometry $\phi$ from $x$ to $y$ has the   closed form of
> $$ \phi = \phi_x \circ \phi_y ,  $$
> 	where $\phi_x(z) =\frac{x-z}{1-\bar x z} $ and $\phi_y(z) =\frac{y-z}{1-\bar yz}$. Therefore, when $K$ has diameter $D$,   we can figure out a bound of $\rho$ in
> $$ \rho \leq 16\frac{1+\tanh(D/2)}{1-\tanh(D/2)} (\frac{1}{1- \tanh(2D)^2 }+ \frac{D}{\tanh(D/2)}), $$
> which indeed may grow exponentially with respect to $D$.
>
> *  An upper  bound on $\frac{S_\delta}{V_\delta}$ can be easily established for all $\delta>0$, which equivalently imply $\epsilon(n)$ can be taken as  $\infty$. The following as our new estimation.
>     There holds
>      $$ \frac{S_\delta}{V_\delta} = \frac{ \sinh^{n-1}(\sqrt{\kappa}\delta)}{\int_0^\delta \sinh^{n-1}(\sqrt{\kappa}t) dt. } $$
>    By a change of variable $u = \sinh t$, we find
>    $$ \int_0^\delta \sinh^{n-1}(\sqrt{\kappa}t) dt = \kappa^{-1/2} \int_0^{\sinh(\sqrt{\kappa}\delta)} u^{n-1} (1+u)^{-1/2} du $$
>     Integration by parts gives
>     $$ \int_0^\delta \sinh^{n-1}(\sqrt{\kappa}t) dt =  \frac{\sinh^n(\sqrt{\kappa}\delta)}{n \sqrt{\kappa} \cosh(\sqrt{\kappa}\delta)}  + \kappa^{-1/2} \int_0^{\sinh{\kappa\delta}} u^{n-1} (1+u)^{-1/2} du  \ge \frac{\sinh^n(\sqrt{\kappa}\delta)}{n \sqrt{\kappa} \cosh(\sqrt{\kappa}\delta)} $$
>     Putting it into the expression of $\frac{S_\delta}{V_\delta}$, we get
>     $$ \frac{S_\delta}{V_\delta} \le n\sqrt{\kappa}\coth(\sqrt{\kappa}\delta).$$
>     Applying the inequality $\coth(x)<x+1/x$, we have
>     $$ \frac{S_\delta}{V_\delta} \le \frac{n}{\delta} + n\kappa\delta, \quad \forall \delta>0. $$
>     Thus,replacing $ B$ with $n\kappa $ in Theorem 4,  essentially we have $\epsilon(n)=\infty$.
> In summary, we believe the constant magnitudes in all our regret bounds are actually clear. We will surely  highlight these magnitudes in the final version of our paper.
>
>
>
>
>      Q3: Step size in the simulation.
>
> In the hyperbolic case, we use the R-BAN algorithm whose step size is actually constant.
>
> In the SPD case, we carry out the experiment with the modified step size $\eta_t = \sqrt{ \frac{\tanh( \sqrt{\kappa} D)}{( \sqrt{\kappa} D t)}}$.
> Moreover, we are currently  evaluating our R-OGD and R-BAN algorithms on the suggested operator scaling problem, which will be included  in the camera-ready version.
>
>     Q4: Online mirror descent.
>
> We thank you for your suggestion for Riemannian mirror descent methods. Online mirror descent (OMD) methods and the follow-the-leader algorithm (as one kind of OMD) are very essential in the Euclidean online optimization. The paper you provided is of great interest, which provides a solid foundation for the design of Riemannian OMD algorithms in the future.

---

> > ### Comment · Reviewer_6h8i · 2021-08-10
> > **Response**
> >
> > I think you are missing a $\sqrt{\kappa}$ in the bound of $\frac{S_\delta}{V_\delta}$, please do not forget about it in the final version.
> >
> > Thanks for the example bounding \rho. Please be sure to mention and highlight the exponential dependence of the bound on the diameter in the main part of the paper. This seems to be natural for these kinds of methods. For instance, for the mirror descent of the paper I mentioned in my other comment "Global Riemannian Acceleration in Hyperbolic and Spherical Spaces", and for the accelerated algorithm in such work, there is also a constant factor that increases exponentially with the diameter in the case the sectional curvature is negative.
> >
> > In view of the authors' response, I am increasing my score by 1, from 5 to 6.

---

### Official Review · Reviewer_Aqzk · 2021-07-05

**Rating:** 7
**Confidence:** 4

**Summary:**

The paper considers the problem of online optimization over manifolds of negative curvature (Hadamard) in the full-information and bandit setting, providing upper bounds for the regret which match the Euclidean case. Prior work has considered the same problem in a zero-order oracle regime and provided only asymptotic results. The theoretical analysis is an adaptation of [40] in the convex case and [18] in the strongly-convex case, using the law of cosines for geodesic triangles on Hadamard manifolds developed in [37] and new geometric results based on the extra assumption of homogeneity. Experimental results in the simple problem of Karcher mean estimation are also included.

**Limitations And Societal Impact:**

The authors discussed some of the limitations of their work, but didn't discuss others, as mentioned in the completeness part above. Also, I worry for the constant $\rho$ which appears in the upper bound inside $\Lambda$ and some care should be taken about it.

There is not direct societal impact.

**Main Review:**

Originality: I find the main idea of the paper to be a reasonable continuation of previous trends in Riemannian Optimization, thus I am not surprised by its originality. Riemannian methods have gain interest during the last years due to applications in important problems relevant to neurips community (e.g. operator scaling). Technically, the work adapts directly theoretical analysis from the Euclidean case taken from previous work, surpassing obstacles occuring by the geometry of manifolds. In the convex case the well-known law of cosines developed in [37] (which has been used in numerous papers in the area) turns to be enough for a simple adaptation, but in the strongly convex case the authors develop a way to compute the gradient of the integral of a volume form exploiting the assumption that the manifold is homogeneous and the existence of Killing vector fields implied by it. They also prove a sub-convexity property for the same "smoothed loss function" and the deviation from convexity turns out not to affect the final regret bound critically. This is because the extra "Riemannian" term appears in the bound as $\lambda T$ and there is anyway a term $2 \alpha c T$ even in the Euclidean case.

Objections:
I think that the proof of commutativity of gradient and integral can still be done in the manifold case and without homogeneity. The integral of a differential form can be written by definition as the sum of pullbacks of the function through different charts of a coordinate system choosing some partition of unity. Here the manifold is of negative curvature, thus the ball $B_{\delta}(x)$ can be covered only be one system of coordinates (geodesically normal coordinates based at point $x$). Thus the covariant derivative at $x$ is going to become covariant derivative at $0$ in the geodesically normal coordinate system (should be easy to check using the definition of derivative with limits) and now you have the derivative of an integral over the lift of $B_{\delta}(x)$ in the tangent space $T_x M$. You can now put the derivative inside the integral, do the chain rule, and use that the differential of geodesically normal coordinates at the origin is the identity and arrive to the result. Do I miss something?

Regarding the sub-convexity result (lemma 2), I again doubt that homogeneity is essential but didn't have time to check in detail. I worry a bit about the constant $\rho$ which is defined as the supremum of the divergence of a vector field over the working geodesically convex domain. At least this constant is $0$ in the Euclidean case because the isometry $\phi$ can be chosen to be the identity thus $V=0$ and the Euclidean case is recovered, still it is pretty unclear how such a constant depends on the curvature and a bound for the domain $\bar {K}$. Could for instance $\rho$ depending exponentially bad on the domain? This is important in my opinion and an answer could be provided by a formula of divergence based on the determinant of the metric and partial derivatives of it, that the authors could find easily on the internet.

Regarding related work, I am not familiar enough with the literature in online learning to have a strong opinion, the paper though explains clearly the difference with [28], which seems to be in the same topic.

Quality: The paper is technically adequate for the area of Riemannian Optimization. As noted above, I think that it introduces unnecessary technicalities and assumptions. I would like to see some intuitive explanation for the necessity of homogeneous assumption and the resulting existence of Killing vector fields, homogeneous manifolds though are already a rich class of practical cases for machine learning tasks. The paper is sufficiently complete, two things that I think are important: i) I don't see the reason that manifolds of positive curvature are not treated, usually optimization behaves even better on positive curvature. Such an algorithm on the sphere (which is homogeneous) could serve as excellent basis for online PCA which is an important problem. ii) I find the experimental section quite poor because it considers only the (in most cases) strongly convex problem of Karcher mean estimation. Another problem to implement on SPD manifold which is just geodesically convex is operator scaling.

Clarity: The paper is in general clearly written, there are parts that are difficult to be followed by readers without strong background in geometry, but this is the case with lots of papers in the area. Below are some minor comments about readability and typos:

Algorithm 1: the step-size $\eta_t$ is missing

Lemma 10: I find the notation $V_r(p)$ misleading, since the volume of the ball of radius $r$ centered at $p$ does not depend on $p$ because of homogeneity. Then $V_r(\kappa)$ is introduced for a similar ball in a model space, I think this notation must be improved.

224: "quadratic"

I don't understand the reason that the step-size is denoted $\eta_t$ in full-information and $\nu$ in bandit (since the last is also time-dependent). I see that the authors adopted the notation from the relevant Euclidean papers, but since both cases are presented in the same paper now, one notation needs to be used.


Significance: The results are added to the big literature of Riemannian algorithms, with significance for neurips community due to its practical applications. The authors comment that do not treat the even more practical case of retraction-based gradient descent which would be useful, but still the exponential map is computable in SPD and hyperbolic space. Overall, I believe that the paper contains non groundbreaking but good to know results towards the correct direction.

Given all the above, I believe that the paper is a good contribution, but has issues that could make a second reviewing round beneficial. Thus in my opinion it is slightly above acceptance.






**Time Spent Reviewing:**

20

---

> ### Author Response · Authors · 2021-08-10
> **Authors' response**
>
> We sincerely thank you for the constructive comments and encouragement. We will correct the typos and notations that you pointed out in the final version.
>
> 	Q1: The necessity of the assumption of homogeneity.
> We appreciate your suggestion about removing the homogeneity. The challenge lies in, deviating the integral in the lifting space $T_xM$ relies on  the Riemannian volume form $\omega = \sqrt{G}d x_1\wedge\dots\wedge d x_n$. That is $$ \nabla_X\hat{f}(x) = \int_{B_\delta(x)} (\nabla_X f \sqrt{G} + f \nabla_X \sqrt{G} )dx_1\wedge\dots\wedge dx_n,$$ which may cause the commutativity to fail to hold. Then homogeneity implies that the derivative of $\hat{f}(x)$ along $X$ can be treated as a derivative along the corresponding Killing field $\eta$, where $ \nabla_\eta \sqrt{G} $ vanishes. As a result, the critical commutativity holds, which allows our analysis to stand on a rigorous footing.
>
>     Q2: Positive curvature manifold are not treated.
>
> Indeed there have been a few frameworks for optimization over manifolds with positive curvature. In the online setting that is considered in the current paper, the regret analysis crucially depends on the non-expansiveness of the projection map (Lemma 3, or Corollary 8 in [37]) over manifolds with non-positive curvature, which no longer holds with positive curvature. Therefore, we limit our discussions on manifolds with non-negative curvature in our current work.
>
>  In the meantime, we would like to note that non-positive curvature manifolds also play important roles in several machine learning applications,  such as the Riemannian dictionary learning on SPD($n$) [1], DTI denoising on SPD($n$) [2], and hyperbolic graph CNN [3].
>
> [1] https://arxiv.org/pdf/1507.02772.pdf
> [2] https://pubmed.ncbi.nlm.nih.gov/27168594
> [3] https://arxiv.org/abs/1910.12933
>
>     Q3: Time independent constant $\rho$
> The constant $\rho$ can be expressed as
> 	$$ \rho = \sup_{x,y,u\in K} |\frac{1}{\sqrt{G}} \frac{\partial{}}{\partial{x_i}}\big(\sqrt{G} \exp^{-1}_u \phi(u)\big)^i| \quad s.t. \quad \phi(x) = y  $$
> 	Hence, for a certain manifold $M$, once the set $K$ is fixed and the explicit expression of $\phi$ is given, we can compute the constant $\rho$ as a finite number.
> 	For example, for a $2$-dimensional Poincaré disk, the isometry $\phi$ from $x$ to $y$ has the closed form of
> 	$$  \phi = \phi_x \circ \phi_y ,  $$
> 	where $\phi_x(z) =\frac{x-z}{1-\bar x z} $ and $\phi_y(z) =\frac{y-z}{1-\bar yz}$. Therefore, when $K$ has diameter $D$,   we can figure out a bound of $\rho$ in
>     	$$ \rho \leq 16\frac{1+\tanh(D/2)}{1-\tanh(D/2)} (\frac{1}{1- \tanh(2D)^2 }+ \frac{D}{\tanh(D/2)}), $$
> which indeed may grow exponentially with respect to $D$.
>
>     Q4: Experiments for opertor scaling
> We are evaluating our R-OGD and R-BAN algorithms on the suggested operator scaling problem, which will be included  in the camera-ready version.

---

> > ### Comment · Reviewer_Aqzk · 2021-08-19
> > **thanks for a convincing response**
> >
> > I am not convinced by the necessity of homogeneity but I understand that it is non trivial to remove it. The exponential dependence of $\rho$ on $D$ seems a bit ugly, but it is the case with other papers in the area, and it happens often in negative curvature because volumes increase exponentially fast. The discussion about $\rho$ and an example should be added in the CR version in case of acceptance. Some new experiments would be ideal.
> >
> > Overall, I think the paper has minor flaws but constitutes a solid contribution, therefore I raise my score by one point.

---

### Official Review · Reviewer_BgPU · 2021-07-16

**Rating:** 7
**Confidence:** 3

**Summary:**

The paper proposes Riemannian variations of online optimization problems. Two algorithms (one in the general case and one for bandits) are proposed for (strongly) geodesically convex functions. Error bounds are derived under various regularity conditions on the manifold. The algorithm is evaluated numerically on computing Karcher means.

**Ethical Concerns:**

No ethical concerns.

**Limitations And Societal Impact:**

The authors have addressed the limitations and potential negative societal impact of their work.

**Main Review:**

Strengths

* The paper is easy to read and follow
* The subtleties when working on Riemannian manifolds are addressed well

Weaknesses

* The applications of online learning on Riemannian manifolds are questionable.

Comments

* I would $\textbf{strongly}$ recommend using Fréchet mean instead Karcher mean in the paper. The two refer to the same thing, but Karcher himself has noted that terminology of "Karcher mean" arose in the 90's and misasigns contribution [1].
* On that note, I would also recommend incorporating some more recent work on the application of Fréchet means [2, 3, 4]
* In particular, the Riemannian batch normalization algorithms of [2, 3] could be of potential interest, as one must maintain a moving average of previous data points as one updates the network.

Verdict

The proposed method is technically sound and the paper is well structured. Outside of my comments, I believe this to be a strong paper. As such, I tend accept.

[1] https://arxiv.org/abs/1407.2087
[2] https://arxiv.org/abs/1909.02414
[3] https://arxiv.org/abs/2003.00335
[4] https://arxiv.org/abs/1111.3120


**Time Spent Reviewing:**

3

---

> ### Author Response · Authors · 2021-08-10
> **Authors' response**
>
> We appreciate your constructive suggestions and positive comments.  We will correct the typos and unclear notations that you pointed out in the final version.
>
>     Q1: Fréchet mean and Riemannian batch optimization.
> In the camera-ready version, we will use the equivalent term "Fréchet mean". Moreover, as you have pointed out, the Riemannian batch normalization (R-BN) method [2,3] has proven to be effective for deep neural network training, where updating the running mean can be viewed as an online Fréchet mean problem as the batch generated sequentially. We will highlight this in the final version, and further explore the connection between the Riemannian batch normalization and our online Riemannian optimization framework in the future.
>
>     Q2: The applications of online learning on Riemannian manifolds are questionable.
> In fact,  there have been a number of machine learning problems that are intrinsically defined over Riemannian manifolds. For example, Ref. [5] (https://arxiv.org/pdf/1507.02772.pdf) proposed a Riemannian dictionary learning approach on SPD($n$) for texture recognition and object identification. When the training data arrive sequentially in such applications, the proposed online Riemannian learning framework may be an ideal solution.

---

> > ### Comment · Reviewer_BgPU · 2021-08-10
> > **Updated Review**
> >
> > Thank you for addressing my concerns. In light of this, I am raising my score from 6 -> 7.

---

### Official Review · Reviewer_7KFW · 2021-07-16

**Rating:** 7
**Confidence:** 4

**Summary:**

In this paper, the authors derived Riemannian online learning algorithms with first-order derivative information on Hadamard manifolds, with various problem settings (including full information geodesically-convex, full information geodesically-strongly-convex and bandit feedback). In each setting, the authors proved the regret of their proposed algorithms matched their vector space counterparts (up to Riemannian geometric constants). In deriving their proofs, the authors introduced new analytic tools and insights to Riemannian optimization research, which could facilitate further developments in this area.

**Limitations And Societal Impact:**

The authors addressed the limitations of their work. No specific potential negative societal impact is introduced by this work.

**Main Review:**

Originality:

-- The Riemannian geometry analysis of Lemma 1 and Lemma 2 seems to be highly original in the Riemannian optimization community. The other parts of their results are original as well, especially since they need to envision using Riemannian geometry lemmas such as Lemma 1 to alleviate challenges blocking a direct generalization of the vector space analysis.

Quality:

-- I find the paper well-written and the results convincing. The authors also carefully reviewed related work. The experiments do not seem exciting from a practical viewpoint, but they are sufficient to support the theoretic results.

Clarity:

-- The paper presents ideas clearly and is easy to follow. Each assumption and claim (lemma / theorem) is preceded or followed by some discussion of its implication, to help the readers navigate through the development.

-- Possible typo: Line 288, should probably be "R-OGD-SC converges faster than that of R-OGD-C ..."

Significance:

-- This paper is the first to generalizes finite iteration regret bound analysis to Riemannian online learning algorithms.

-- Riemannian geometry lemmas such as Lemma 1 and Lemma 2 provide new analytic tools to analyze optimization and learning problems on Riemannian manifolds.

**Time Spent Reviewing:**

4

---

> ### Author Response · Authors · 2021-08-10
> **Authors' response**
>
> Thank you for your positive comments. We will correct the typo in Line 288 that you pointed out in the final version in order to improve the quality of the manuscript.
>
> Indeed, in this paper, to construct no-regret learning algorithms on Riemannian manifolds, we had to rely on several deep knowledge in Riemannian geometry and establish several effective techniques to surmount the manifold geometry. For instance, we construct the online Riemannian Busemann optimization with Busemann functions to prove a universal regret lower bound; and we manage to estimate a gradient estimator on Riemannian manifolds via Killing fields and the Bishop-Gromov volume comparison theorem. These knowledge and techniques allow us to build solid understanding of online optimization over Riemannian manifolds as presented in the current paper, and may shed lights on more machine learning problems on Riemannian manifolds in the future.

---

### Decision · Program_Chairs · 2021-09-27

**Decision:**

Accept (Poster)

**Comment:**

This paper considers online optimization problems in Hadamard manifolds - i.e., simply connected, complete Riemannian manifolds with everywhere non-positive sectional curvature. The authors consider several different settings - full-gradient versus gradient-free algorithms against both convex or strongly-convex objectives - and they provide bounds that mirror the corresponding bounds for the Euclidean case.

The reviews speak for themselves and the concerns raised during the review phase were addressed by the authors. As a result, there were no reservations about making an "accept" recommendation.

In preparing the camera-ready version of their paper, the authors should make sure to include all the comments made by the reviewers. In addition, I would have the following recommendations/questions:
- An aspect which has regrettably been overlooked in the paper is what the Hadamard requirement really means. By the Cartan-Hadamard theorem, every $n$-dimensional Hadamard manifold is diffeomorphic to $\mathbb{R}^n$, so the topology of these manifolds is trivial. This is a crucial limitation, because barycentric constructions cannot be defined otherwise (at the very least, not easily), and it is not possible to use the theory of the paper to solve optimization problems defined on, say, a matrix group – like the set of invertible matrices $\mathrm{GL}(n)$, or the set of orthogonal matrices $\mathrm{O}(n)$. The paper doesn't make any such allusions but, at the same time, it's important to state clearly - and early - that Hadamard manifolds are topologically (and diffeomorphically) trivial.
- The section on Riemannian manifolds should be expanded, and the authors should be careful with the notation they employ: I am strongly in favor of representing vector fields as differential operators (which they do, sometimes explicitly other times implicitly), but this should be carefully explained, and some elements of the appendix should be transferred to the main text. In this regard, the authors might find helpful Lee's textbooks on smooth and Riemannian manifolds - the notation is already quite close, so this would essentially be a matter of fixing a few glitches, as in the definition of the Hessian.
- I would also be curious to see a comparison between the authors' work and recent Riemannian approaches to online optimization like the 2019 ICLR paper by Bécigneul and Ganea ("Riemannian adaptive optimization methods") and the 2020 ICLR paper by Antonakopoulos et al. ("Online and Stochastic Optimization beyond Lipschitz Continuity: A Riemannian Approach"). A more detailed presentation of previous works on the topic - like Bonnabel's (2013) paper - would also help with the positioning of this work. [To be clear, the settings and results are quite different and there is no issue of an overlap, but explaining these differences would be helpful to the reader]